# Measurement of the axial vector form factor from antineutrino–proton scattering

T. Cai[1,2✉], M. L. Moore[1,26], A. Olivier[1], S. Akhter[3], Z. Ahmad Dar[3,4], V. Ansari[3], M. V. Ascencio[5,27], A. Bashyal[6,28], A. Bercellie[1], M. Betancourt[7], A. Bodek[1], J. L. Bonilla[8], A. Bravar[9], H. Budd[1], G. Caceres[10,29], M. F. Carneiro[6,10,30], G. A. Díaz[1], H. da Motta[10], J. Felix[8], L. Fields[11], A. Filkins[4], R. Fine[1,31], A. M. Gago[5], H. Gallagher[12], S. M. Gilligan[6], R. Gran[13], E. Granados[8], D. A. Harris[2,7], S. Henry[1], D. Jena[7], S. Jena[14], J. Kleykamp[1,32], A. Klustová[15], M. Kordosky[4], D. Last[16], T. Le[12], A. Lozano[10], X.-G. Lu[17,18], E. Maher[19], S. Manly[1], W. A. Mann[12], C. Mauger[16], K. S. McFarland[1✉], B. Messerly[20,33], J. Miller[21], O. Moreno[4,8], J. G. Morfín[7], D. Naples[20], J. K. Nelson[4], C. Nguyen[22], V. Paolone[20], G. N. Perdue[1,7], K.-J. Plows[18], M. A. Ramírez[8,16], R. D. Ransome[23], H. Ray[22], D. Ruterbories[1], H. Schellman[6], C. J. Solano Salinas[24], H. Su[20], M. Sultana[1], V. S. Syrotenko[12], E. Valencia[4,8], N. H. Vaughan[6], A. V. Waldron[15,25], M. O. Wascko[15], C. Wret[1], B. Yaeggy[21,34] & L. Zazueta[4]

Scattering of high energy particles from nucleons probes their structure, as was done in the experiments that established the non-zero size of the proton using electron beams[1]. The use of charged leptons as scattering probes enables measuring the distribution of electric charges, which is encoded in the vector form factors of the nucleon[2]. Scattering weakly interacting neutrinos gives the opportunity to measure both vector and axial vector form factors of the nucleon, providing an additional, complementary probe of their structure. The nucleon transition axial form factor, $F_A$, can be measured from neutrino scattering from free nucleons, $\nu_\mu n \to \mu^- p$ and $\bar{\nu}_\mu p \to \mu^+ n$, as a function of the negative four-momentum transfer squared ($Q^2$). Up to now, $F_A(Q^2)$ has been extracted from the bound nucleons in neutrino–deuterium scattering[3–9], which requires uncertain nuclear corrections[10]. Here we report the first high-statistics measurement, to our knowledge, of the $\bar{\nu}_\mu p \to \mu^+ n$ cross-section from the hydrogen atom, using the plastic scintillator target of the MINERvA[11] experiment, extracting $F_A$ from free proton targets and measuring the nucleon axial charge radius, $r_A$, to be 0.73 ± 0.17 fm. The antineutrino–hydrogen scattering presented here can access the axial form factor without the need for nuclear theory corrections, and enables direct comparisons with the increasingly precise lattice quantum chromodynamics computations[12–15]. Finally, the tools developed for this analysis and the result presented are substantial advancements in our capabilities to understand the nucleon structure in the weak sector, and also help the current and future neutrino oscillation experiments[16–20] to better constrain neutrino interaction models.

Form factors measured in scattering processes describe the structure of composite objects. They have been thought to be the Fourier transform of charge distributions in the non-relativistic limit of low negative four-momentum transfer squared, $Q^2$. Although this interpretation is not strictly true[21], the slopes of the form factors at $Q^2 = 0$ provide a measure of the mean-squared radius $\langle r_A^2 \rangle$ for the particle in the charge species described. Nucleon electric ($G_E^N$) and magnetic ($G_M^N$) form factors are precisely measured in electron–nucleon elastic scattering experiments, enabling the radius of the nucleon[22,23] to be inferred. Neutrino scattering measurements yield the analogous axial vector form factor, $F_A$, which characterizes the weak charge distribution. $F_A$ is also a key input to neutrino oscillation experiments to precisely measure the neutrino

oscillation parameters, including CP violation, and to establish mass hierarchy.

Previous measurements of $F_A$ in neutrino scattering were performed by measuring $d\sigma/dQ^2$ in the reaction $\nu_\mu D \to \mu^- pp$ in deuterium bubble chambers[3–9]. Even in deuterium nuclei, theoretical assumptions[10] about the Fermi motion of the bound nucleons, the application of the Pauli exclusion principle in the proton-proton ($pp$) final state and the nuclear wave function are required to extract $F_A$ from these measurements[24].

Previous extractions assumed $F_A$ to follow the dipole form factor, although more flexible models for the form factors are also available[24–26]. Although there are many efforts[12,13,15] to calculate $F_A$ for $Q^2 > 0$ from

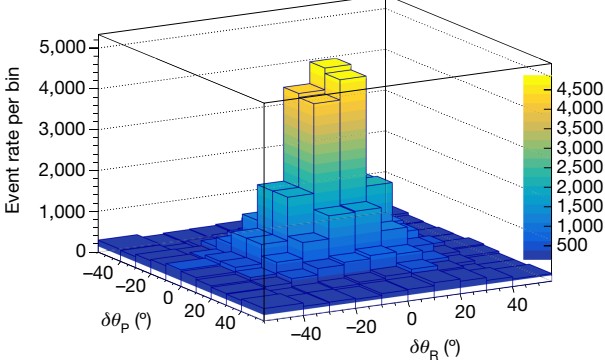

**Fig. 1 | Data rate and the predicted fractional interaction types in the angular plane.** Left, Data event rate in $\delta\theta_P$–$\delta\theta_R$ plane. The visible bins in positive $\delta\theta_P$ are representative of the event rate as the event rate is nearly symmetrical in $\delta\theta_P$. Right, The projected Monte Carlo event fraction in different angular regions. CCE hydrogen (pink), QELike CCQE carbon (green), QELike other (2p2h, resonant, yellow and the non-QELike background (grey) all appear in the event selection. The CCE signal region is between

$-10° < \delta\theta_R/\delta\theta_P < 10°$. Control regions are defined to measure events in regions where QE contributions, non-QE contributions and a mixture of non-QE and meson events are dominant. Two validation regions for the QE and non-QE contributions are defined to assess the background estimations. The background levels shown are constrained by the fit to the data as described in the text.

lattice quantum chromodynamics (QCD) with increasing precision, calculations in the $Q^2$ region above 1 GeV$^2$ remain imprecise. Pion electroproduction[27] and muon capture measurements on hydrogen[28] may also be interpreted as constraints on $F_A$, but again with theoretical uncertainties.

In the charged-current elastic (CCE) reaction on the free nucleon $\bar{\nu}_\mu H \rightarrow \mu^+ n$, a muon antineutrino elastically scatters off the free proton from the hydrogen atom, turning the neutrino into the more massive positively charged muon $\mu^+$ and the proton into a neutron (see section 'Terminology of the signal process'). This reaction is free from the nuclear theory corrections described above in scattering from deuterium (D) and provides a direct measurement of $F_A$. It is also a two-body reaction with a nucleon at rest; therefore, the neutrino direction and the final-state $\mu^+$ momentum fully specify the interacting system. The antineutrino energy ($E_{\bar{\nu}}$) and $Q^2$ are reconstructed under the CCE hypothesis ($Q^2_{QE}$) of equation (1) and use only the muon energy ($E_\mu$), momentum ($p_\mu$) and angle ($\theta_\mu$) of the muon with the neutrino beam. The free proton in the initial state, and the neutron and muon in the final states have masses denoted as $M_p$, $M_n$ and $m_\mu$, respectively.

$$E_{\bar{\nu}} = \frac{M_n^2 - M_p^2 - m_\mu^2 + 2M_p E_\mu}{2(M_p - E_\mu + p_\mu \cos\theta_\mu)}, \tag{1}$$

$$Q^2_{QE} = 2E_{\bar{\nu}}(E_\mu - p_\mu \cos\theta_\mu) - m_\mu^2,$$

The neutron momentum in components parallel and transverse to the neutrino direction are

$$p_n^\| = \frac{M_n^2 + p_\mu^{\perp 2} - (p_\mu^\| - E_\mu + M_p)^2}{2(p_\mu^\| - E_\mu + M_p)^2}, \tag{2}$$

$$\mathbf{p}_n^\perp = -\mathbf{p}_\mu^\perp. \tag{3}$$

The neutron in non-CCE background reactions will deviate from the predicted neutron directions due to nuclear effects, different initial state assumptions and final-state mass, making available physics-driven selections to reduce the backgrounds. No statistically significant measurement of the process has been performed so far; the only measurement previous to this work recorded 13 events in a hydrogen bubble chamber[29].

The free nucleon cross-section is given by

$$\frac{d\sigma}{dQ^2}\begin{pmatrix} \nu n \rightarrow l^- p \\ \bar{\nu} p \rightarrow l^+ n \end{pmatrix} = \frac{M^2 G_F^2 \cos^2\theta_c}{8\pi E_\nu^2}$$
$$\left[ A(Q^2) \mp B(Q^2)\frac{(s-u)}{M^2} + C(Q^2)\frac{(s-u)^2}{M^4} \right], \tag{4}$$

$$A(Q^2) = \frac{m^2 + Q^2}{4M^2}\left[ \left(4 + \frac{Q^2}{M^2}\right)|F_A|^2 - \left(4 - \frac{Q^2}{M^2}\right)|F_V^1|^2 \right.$$
$$\left. + \frac{Q^2}{M^2}\left(1 - \frac{Q^2}{4M^2}\right)|\xi F_V^2|^2 + \frac{4Q^2}{M^2}F_V^1\xi F_V^2 + \mathcal{O}\left(\frac{m^2}{M^2}\right) \right], \tag{5}$$

$$B(Q^2) = \frac{Q^2}{M^2}F_A(F_V^1 + \xi F_V^2),$$

$$C(Q^2) = \frac{1}{4}\left(|F_A|^2 + |F_V^1|^2 + \frac{Q^2}{4M^2}|\xi F_V^2|^2\right)$$

where $M = (M_n + M_p)/2$ is the average nucleon mass, $G_F$ is the Fermi coupling constant, $\theta_c$ is the Cabibbo angle, $E_\nu$ is the neutrino energy, $(s-u) = 4ME_\nu - m^2 - Q^2$ and $m$ is the charged lepton mass[30]. The parameters $A$, $B$ and $C$ are functions of $F_A$, the two vector form factors $F_V^1$ and $\xi F_V^2$ derived from the proton and neutron electric and magnetic form factors, and a pseudoscalar form factor $F_P$ whose effect is suppressed by a factor of $m^2/M^2$ in $A$. The form factors are real assuming T invariance[30] and charge symmetry. The vector form factors have been precisely parameterized from results of electron scattering experiments[25,26,31,32]. The pseudoscalar form factor is predicted from the axial form factor[33]. The cross-section reported in this paper is a convolution between the free nucleon cross-section and the wide-band neutrino flux[34], with restrictions on the muon kinematics due to the detector geometry. $F_A$ can be derived from the restricted and flux-convolved cross-section given the other form factors.

The nucleon axial radius may be found from the slope of the small $Q^2$ expansion of $F_A$ at $Q^2 = 0$:

$$F_A(Q^2) = F_A(0)\left(1 - \frac{\langle r_A^2\rangle}{3!}Q^2 + \frac{\langle r_A^4\rangle}{5!}Q^4 + ...\right),$$
$$\frac{1}{F_A(0)}\frac{dF_A}{dQ^2}\bigg|_{Q^2=0} = -\frac{1}{6}\langle r_A^2\rangle \tag{6}$$

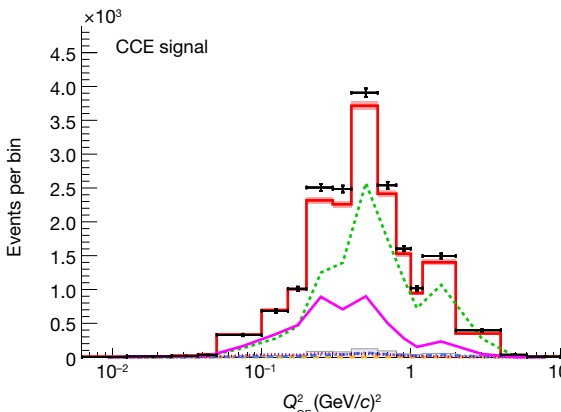
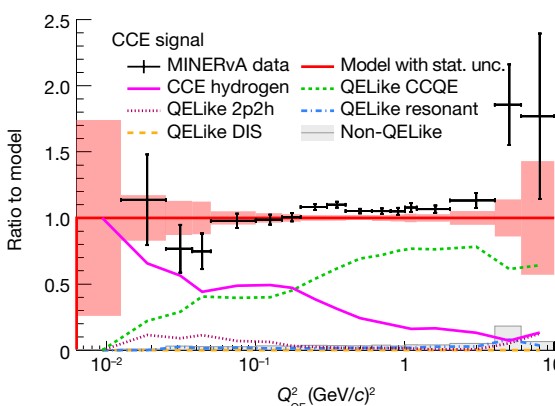

**Fig. 2 | Fitted event distribution and ratio in the signal region.** Event rate (left) and ratio to the post-fit model (right). The vertical error bars around the data points and the error band around the model prediction account for statistical uncertainty (stat. unc.) based on the standard deviation of a Poisson distribution. Note that the $(0, 0.0125)$ $(GeV/c)^2$ and $(6.0, 10)$ $(GeV/c)^2$ bins are not reported due to low statistics. DIS refers to deep inelastic scattering.

where $F_A(0) = -1.2723 \pm 0.0023$ is derived from neutron decay measurements[35]. We define $r_A \equiv \sqrt{\langle r_A^2 \rangle}$.

## Experiment description

The CCE process on hydrogen was measured with the MINERvA[11] detector in a $\bar{\nu}_\mu$ beam produced at the NuMI neutrino beamline[34] at Fermilab with an average energy of 5.4 GeV. Antineutrino interactions are selected by requiring a $\mu^+$ and detected neutron signatures in the MINERvA detector.

MINERvA is a segmented scintillator detector with hexagonal planes constructed from strips of triangular cross-section assembled into planes perpendicular to the $\bar{\nu}_\mu$ beam, and is described in more detail in the Methods. This analysis reconstructs neutrino interactions in the active tracker region of the detector, which consists of the scintillator. The scintillator strips point in either the vertical ($X$) or one of the $\pm 60°$ ($U$, $V$) directions. This region is fully active, consisting of 128 tracker planes stacked in alternating patterns of $XUXV$. The alternating orientation enables extraction of a three-dimensional position from the strips when charged particles traverse two or more consecutive planes. Muons produced from charged-current neutrino interactions in the MINERvA detector may exit from the rear and enter the MINOS near detector (ND)[36], which is located immediately downstream of the MINERvA detector. The MINOS ND is a fully magnetized scintillator and steel detector that determines the muon's charge and momentum by measuring its curvature and range.

Only muons in the energy range 1.5 GeV < $E_\mu$ < 20 GeV with an opening angle $\theta_\mu < 20°$ with respect to the neutrino direction are selected because they can be efficiently measured by the MINOS ND. The vertex is defined to be the beginning of the muon track. Energy deposits from other charged particles, such as protons and $\pi^\pm$, can be reconstructed into tracks if they span at least four planes. Photon pairs from $\pi^0$s can be reconstructed from their electromagnetic showers.

Although neutrons are not directly observable from ionization as charged particles, they produce secondary particles with observable energy deposits when they elastically, quasi-elastically or inelastically scatter in the detector. The dominant interactions produce low-energy protons, which can be observed[37]. Neutrons also scatter undetectably, for example by inelastically knocking out neutrons from carbon nuclei or elastically scattering from carbon, in ways that change the neutron direction and energy. Monte Carlo simulation studies of single-neutron transport in the MINERvA detector show that the angle between the reconstructed and true neutron directions follows the sum of two exponential distributions, with 68% of the candidates within 12°.

## Cross-section extraction

The CCE cross-section is measured in bins of $Q^2$. Control samples, events with neutrons pointing away from the predicted direction, provide data-driven constraints on the background models as a function of $Q^2$. The reconstructed neutron directions from the signal events centre around the predicted direction from the $\mu^+$ reconstruction with deviations due to the angular resolution. Additional nuclear effects alter the initial neutron directions when the neutron is produced by the charged-current quasi-elastic (CCQE) neutrino interaction on a bound proton in the carbon nucleus. Fermi motion in the carbon nucleus imparts each bound nucleon with a random initial momentum resulting in the neutron direction further deviating from the two-body calculation. Although the CCQE cross-section is a function of $F_A$ as well, measurements of electromagnetic form factors even in nuclei as light as $^4$He have shown that nuclear effects obscure the relationship and make extractions of $F_A$ from measurements CCQE on $^{12}$C (refs. [38–40]) susceptible to uncertain nuclear physics. Additionally, multinucleon knockout, such as two-particle-two-hole (2p2h) reactions, and secondary interactions of outgoing neutrons before exiting the nucleus are also present in carbon. The latter phenomena, collectively termed the final-state interactions (FSI), can change the direction and energy of the neutron, and can produce additional final-state particles, including pions produced through the excitation and subsequent decay of nucleons in the nucleus.

On the basis of the detected final-state particles, events are divided into those with only nucleons in the final state (QELike) and those with mesons present (non-QELike). Although the CCE signal is an exclusive subset of QELike, the carbon CCQE, 2p2h and resonant pion production events may experience FSI and land in either category. Both the signal and the background processes are simulated using a realistic Monte Carlo simulation of the detector based on the GEANT4 (ref. [41]) simulation toolkit. The input model for neutrino-nucleus interactions is based on GENIE[42] with theory and data-driven modifications[43–48]. Finally, the Fermi gas initial state model used by GENIE has been reweighted into a spectral function (SF)[49,50] for a more realistic description of nucleus initial states[51].

Samples with predominantly QELike(non-QELike) events are selected by requiring ≤1(>1) energetic shower(s) in the detector. Each sample is subdivided according to the neutron candidates' opening angles to

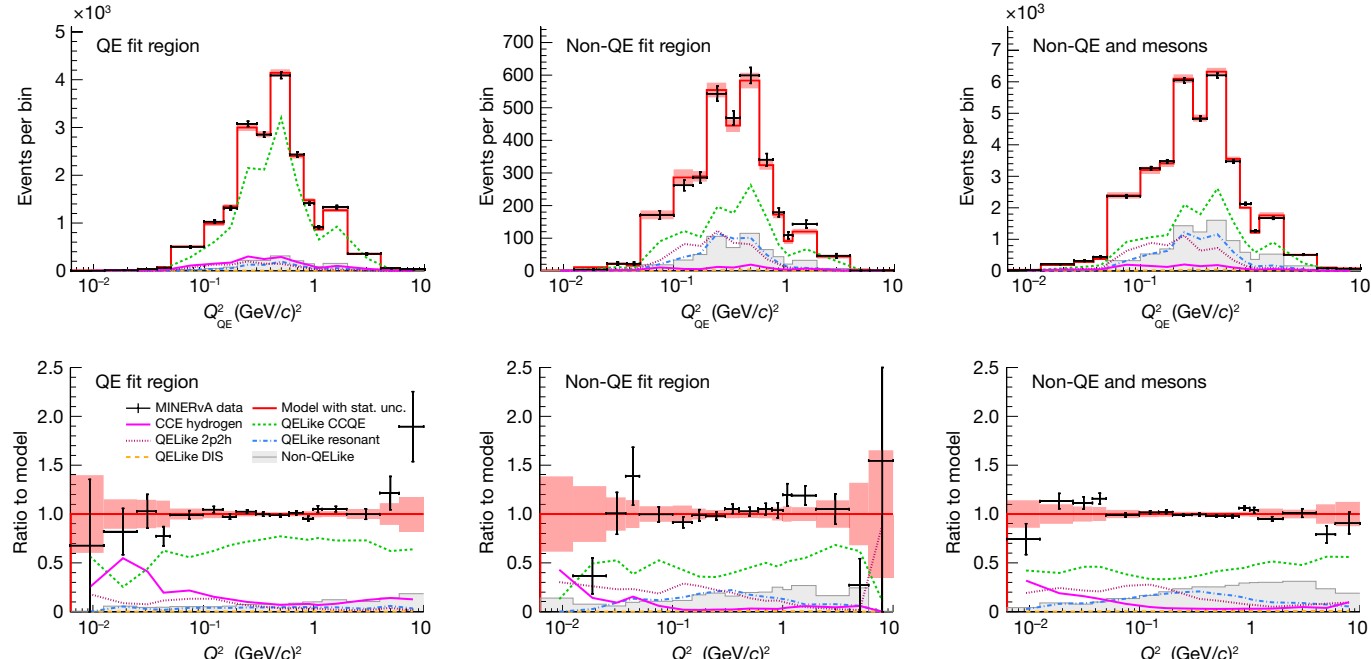

**Fig. 3 | Fitted event distributions and ratios in the control regions.** Fitted event distribution in the calculated $Q^2$ ($Q^2_{QE}$) bins in the QELike sample in the QE, non-QE, as well as non-QE and mesons, regions. Top, event rate per bin; Bottom, ratio to post-fit model. The vertical error bars around the data points and the error band around the model prediction account for 1 standard deviation due to large statistical uncertainties.

the predicted neutron direction on hydrogen to constrain the QELike and non-QELike backgrounds due to carbon. An alternative coordinate system can be defined as follows:

$$\hat{z}' = \hat{t}, \ \hat{x}' = \hat{p}_\nu \times \hat{p}_\mu, \ \hat{y}' = \hat{z}' \times \hat{x}',$$
$$\delta\theta_P = \arctan((\hat{n} \cdot \hat{x}')/(\hat{n} \cdot \hat{z}')),$$
$$\delta\theta_R = \arctan((\hat{n} \cdot \hat{y}')/(\hat{n} \cdot \hat{z}')),$$

(7)

where the predicted neutron direction $\hat{t}$, together with the neutrino direction $\hat{p}_\nu$ and measured muon direction $\hat{p}_\mu$, set up an alternative coordinate system $(\hat{x}', \hat{y}', \hat{z}')$. The angular variables $\delta\theta_R$ and $\delta\theta_P$ describe the projected angles between the measured neutron candidate direction $\hat{n}$ and $\hat{t}$ in the neutrino-muon reaction plane and the orthogonal plane intersecting $\hat{z}'$.

Figure 1 shows the relative contributions of the signal CCE and various backgrounds in each $\delta\theta_P$–$\delta\theta_R$ bin in the QELike sample. Although most neutrons from the CCE interaction are reconstructed in the signal region, some undergo rescattering before they are detected, and are measured in non-signal regions. CCQE interaction is the dominant background in the signal region and the regions immediately surrounding it. Fermi momentum and FSI give CCQE events a broader distribution across $\delta\theta_R$ and $\delta\theta_P$. Other interactions have a much broader distribution and are the main background at large $\delta\theta_P$ and $\delta\theta_R$, but still have a small presence in the signal region.

To isolate CCE interactions, the background events are subtracted on the basis of a Monte Carlo simulation. The angular distributions are predicted by the Monte Carlo, but the data determine the $Q^2$-dependent normalization. The normalization is determined by fitting the 'QE' region, which is dominated by CCQE, together with the 'non-QE' and the 'non-QE and mesons' regions, which have a large fraction of non-CCQE interactions. The normalization as a function of $Q^2$ of each interaction channel is adjusted to provide the best fit for the three regions. The normalizations are validated by comparing to event rates in the 'QE validation' and 'non-QE validation' regions, which have different admixtures of each process. This comparison has an acceptable

$\chi^2 = 13.5$ for 9 degrees of freedom. The normalizations are then extrapolated to the signal region, and the excess of events is attributed to CCE scattering. By using the data to normalize the background, any modifications in the CCQE cross-section due to changes in $F_A$ from the default model are taken into account. A cross-check of the fit strategy was performed on a proton sample in the beam of neutrinos rather than antineutrinos. Reactions of neutrinos are free of CCE events. The strategy yields good agreement at $Q^2 > 0.2$ GeV/$c^2$, the threshold above which CCQE proton tracks can be reconstructed by MINERvA. The antineutrino sample, on the other hand, reaches lower $Q^2$ because the energy deposited by neutron secondary interactions need only span two planes. Figure 2 shows the antineutrino data with the signal and the predicted background levels after the fit, and the ratio to the post-fit model, as a function of $Q^2$ computed under the hydrogen hypothesis ($Q^2_{QE}$) in the signal region. Figure 3 shows the same distributions in the fit regions. CCQE is the dominant background in the signal and QE fit regions.

The effect of detector resolution on the signal events is corrected using an iterative unfolding method[52,53], and the detector efficiency under the muon phase space cut in each bin is assessed using the simulation. The cross-section is calculated using the fully integrated antineutrino flux[54–56] and the total number of hydrogen atoms in the tracker. This measurement does not correct the efficiency loss as a result of the muon phase space cut, which manifests as a restriction on the range of antineutrino energies allowed into the signal sample at each $Q^2$. Theoretical cross-section predictions need to account for the restricted energy range for comparison with the measurement.

## Discussion

This is the first statistically significant measurement, to our knowledge, of the antineutrino CCE scattering on the free proton. We observe 5,580(180) signal events over the estimated background of 12,500 events. The measured cross-section is shown on the left in Fig. 4 in black data points, as a ratio to the cross-section prediction assuming a dipole $F_A$ described below. The analysis is dominated by statistical

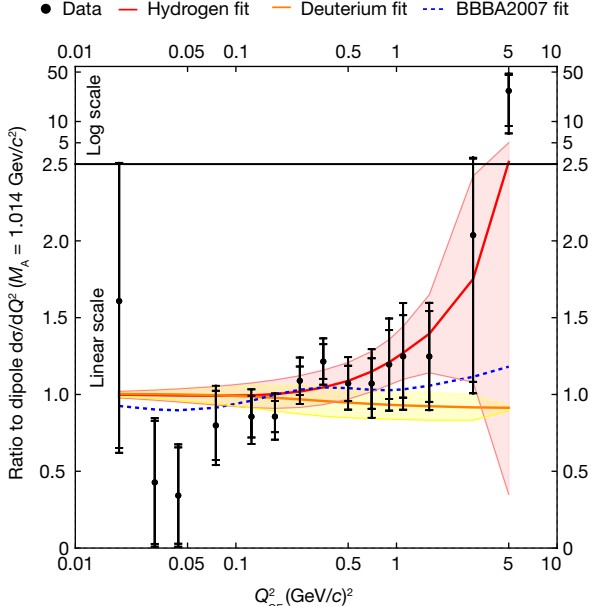

**Fig. 4 | Ratios of data and fitted axial vector form factor to a dipole model.**
Left, ratios of cross-sections to dipole cross-section with $M_A = 1.014$ GeV/$c^2$.
The inner error bars on the data points account for 1 standard deviation due to
statistical uncertainty only, and the full error bars include all sources of
systematic uncertainties. Right, ratios to the dipole form factor. The hydrogen
(this work) and deuterium[24] $F_A$ fits use the $z$ expansion formalism; BBBA2007
(ref. [25]) uses a different empirical fit to deuterium and $\pi$-electroproduction
data; whereas LQCD is a recent fit to lattice QCD calculations[14].

uncertainties at all $Q^2$. Systematic uncertainties arise from the small
remaining differences, due in part to the regularization, between the
post-fit background prediction in each systematic variation of the
input model. The dominant systematic uncertainties in this measure-
ment are the neutron secondary interaction in the detector (4.8%), the
normalization in the CCQE cross-section (4.5%), the muon energy scale
(4.2% from MINOS and 3.1% from MINERvA), the flux (3.9%), neutron
FSI (approximately 3%) and the 2p2h process (2.3%).

Theory prediction of the measured cross-section requires input
from the electromagnetic vector form factors, the axial form factor,
the muon momentum and angle restrictions described above, and
convolution between the free nucleon cross-section with the anti-
neutrino flux. The electromagnetic form factor used in this study
assumes the BBBA2005 (ref. [31]) parameterization. The axial form fac-
tor used by most neutrino experiments and generators[36,42,49,57,58]
assumes a dipole form, $F_A(Q^2) = F_A(0)(1 + Q^2/M_A^2)^{-2}$, which is an approx-
imation derived from the Fourier transform of an exponential charge
distribution. In this ansatz, the shape of $F_A$ depends only on the axial
mass term $M_A$. A more general form, consistent with QCD, is the $z$
expansion formalism[59], which maps the one-dimensional variable
$t = -Q^2$ onto a unit circle bounded by $t_{cut} = 9m_\pi^2$, the threshold of
three-pion production allowed by the axial current[24]:

$$z(Q^2, t_{cut}, t_0) = \frac{\sqrt{t_{cut} + Q^2} - \sqrt{t_{cut} - t_0}}{\sqrt{t_{cut} + Q^2} + \sqrt{t_{cut} - t_0}}$$

$$F_A(Q^2) = \sum_{k=0}^{k_{max}} a_k z^k \qquad (8)$$

The hydrogen cross-section is fitted using $F_A$ from the $z$ expansion
with $t_0 = -0.75$ (GeV/$c$)$^2$, $k_{max} = 8$. $t_0$ is chosen so that the $Q^2$ bins with
precise cross-section measurements are distributed symmetrically
around $z = 0$. Small variations in $t_0$ have no impact on the fit result. $k_{max}$
was chosen to be as small as possible while still enabling the fit to
describe the data, as tested by a $\chi^2$ statistic. The fit to data includes a
bound on the higher order terms[24], such that $|a_k/a_0| \lesssim 5$ and, for $k > 5$,
$|a_k/a_0| \lesssim 25/k$. This bound is treated as a Gaussian regularization term
during the $\chi^2$ minimization process with a strength parameter $\lambda$.
The optimal $\lambda$ of 0.13 was determined by an L-curve study comparing
the minimum $\chi^2$ separated into the comparison to the data and
the regularization. The behaviour of $F_A$ at low $Q^2$ is constrained by
$F_A(0) = -1.2723 \pm 0.0023$, the axial vector coupling as measured in beta
decay. A more detailed discussion of the fitting method can be found
in the Methods.

The resulting cross-section fit (in red) is shown on the left of Fig. 4 as
ratio to a predicted dipole cross-section with $M_A = 1.014$ GeV/$c^2$, together
with the predicted cross-section using $F_A$ from the Meyer[24] fit (in yellow)
on deuterium data and a fit derived jointly from deuterium and pion
electroproduction data (BBBA2007, in dotted blue)[25]. The resulting
form factor as a ratio to the dipole form factor is shown on the right.
The cross-section ratio scales approximately linearly with $F_A$ ratios due
to suppression of the $A$ term in equations (4) and (5). The nucleon axial
radius from the fit to this result is $r_A \equiv \sqrt{\langle r_A^2 \rangle} = 0.73(17)$ fm.

This result is the first statistically significant measurement, as far as
we are aware, of the axial vector form factor on free protons without
nuclear corrections or other theoretical assumptions. Theoretical
uncertainties from the carbon background have been minimized by
data-driven methods. By providing a precise and reliable prediction
for the charged-current elastic scattering from nucleons, neutrino
measurements on higher $Z$ nuclei can benefit from better constrained
nucleon effects to expose the nuclear effects. The method developed
in this study will enable future experiments with hydrogen content in
the target[18,19] to make further measurements of the axial form factor.
Future experiments with intrinsic three-dimensional capability would
be able to observe the directions of low-energy neutron candidates,
and improve the low $Q^2$ measurement with more statistics.

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

[1]Department of Physics and Astronomy, University of Rochester, Rochester, NY, USA. [2]Department of Physics and Astronomy, York University, Toronto, Ontario, Canada. [3]AMU Campus, Aligarh, India. [4]Department of Physics, William & Mary, Williamsburg, VA, USA. [5]Sección Física, Departamento de Ciencias, Pontificia Universidad Católica del Perú, Lima, Peru. [6]Department of Physics, Oregon State University, Corvallis, OR, USA. [7]Fermi National Accelerator Laboratory, Batavia, IL, USA. [8]Campus León y Campus Guanajuato, Universidad de Guanajuato, Guanajuato, Mexico. [9]University of Geneva, Geneva, Switzerland. [10]Centro Brasileiro de Pesquisas Físicas, Rio de Janeiro, Brazil. [11]Department of Physics, University of Notre Dame, Notre Dame, IN, USA. [12]Physics Department, Tufts University, Medford, MA, USA. [13]Department of Physics, University of Minnesota – Duluth, Duluth, MN, USA. [14]Department of Physical Sciences, IISER Mohali, Mohali, India. [15]The Blackett Laboratory, Imperial College London, London, UK. [16]Department of Physics and Astronomy, University of Pennsylvania, Philadelphia, PA, USA. [17]Department of Physics, University of Warwick, Coventry, UK. [18]Department of Physics, University of Oxford, Oxford, UK. [19]Massachusetts College of Liberal Arts, North Adams, MA, USA. [20]Department of Physics and Astronomy, University of Pittsburgh, Pittsburgh, PA, USA. [21]Departamento de Física, Universidad Técnica Federico Santa María, Valparaíso, Chile. [22]Department of Physics, University of Florida, Gainesville, FL, USA. [23]Rutgers, The State University of New Jersey, Piscataway, NJ, USA. [24]Facultad de Ciencias, Universidad Nacional de Ingeniería, Lima, Peru. [25]School of Physical and Chemical Sciences, Queen Mary University of London, London, United Kingdom. [26]Present address: Department of Physics, Stanford University, Stanford, CA, USA. [27]Present address: Iowa State University, Ames, IA, USA. [28]Present address: High Energy Physics/Center for Computational Excellence Department, Argonne National Laboratory, Lemont, IL, USA. [29]Present address: Department of Physics and Astronomy, University of California at Davis, Davis, CA, USA. [30]Present address: Brookhaven National Laboratory, Upton, NY, USA. [31]Present address: Los Alamos National Laboratory, Los Alamos, NM, USA. [32]Present address: Department of Physics and Astronomy, University of Mississippi, Oxford, MS, USA. [33]Present address: University of Minnesota, Minneapolis, MN, USA. [34]Present address: Department of Physics, University of Cincinnati, Cincinnati, OH, USA. ✉e-mail: tejinc@yorku.ca; kevin@rochester.edu

## Methods

### Terminology of the signal process

This article refers to the $\bar{\nu} + H \to \mu^+ n$ scattering process as the CCE process to distinguish it from the CCQE process that experiments have measured in scattering from bound nucleons in nuclei. The signal process is an isolated, two-body reaction that conserves the invariant mass, and is 'elastic' in the sense that all energy from the neutrino and the target proton is transferred to the final-state muon and neutron.

### Detector and neutrino beam

The MINERvA[11] detector, shown in Extended Data Fig. 1, is located just upstream of the MINOS ND[36] in the NuMI beamline[34] at Fermilab. The main injector supplies a beam of 120 GeV protons striking a graphite target 1.04 km upstream of the detector. Secondary mesons produced at the target are charge selected by two magnetic horns to enhance the $\nu_\mu$ production in the forward horn current mode and the $\bar{\nu}_\mu$ production in the reverse horn current (RHC) mode. The resulting neutrino or antineutrino beam travels at a downward angle of 0.059 radian and has a broad energy profile. In particular, the antineutrino beam in the RHC mode peaks at 5.5 GeV (ref. [60]). Extended Data Fig. 8 (right) shows the flux distribution.

Extended Data Fig. 2 describes the detector components. The MINERvA detector is capable of measuring $\bar{\nu}p \to \mu^+ n$ events over a large range of $Q^2$, up to 7 $(\text{GeV}/c)^2$ by tracking the muon through the scintillator target and reconstructing the secondary interaction of the produced neutrons. Each plane in the active tracker region consists of scintillator strips with a triangular 33 mm (transverse) by 17 mm (along beam) cross-section interlocked together. Particles typically deposit energy in multiple adjacent strips, which gives position resolution smaller than the size of the strips. To be reconstructed for this analysis, muons produced in the scintillator target must exit from the back of the MINERvA detector and be measured by the MINOS ND[36]. The magnetized steel in MINOS provides an average 1.3 T field, and the interspersed scintillator planes measure the length and curvature of the muon tracks to determine the muon's charge and momentum as it enters the MINOS ND. Muon tracks in MINERvA are reconstructed and fitted with a Kalman filter resulting in 3.1 mm tracking resolution. The total muon energy and angle resolutions are 6% and 0.06°, respectively[61]. Muons well tracked by the MINOS ND usually have accepted angles to the beam $\theta_\mu < 20°$ and momenta in the range 1.5 GeV/$c < p_\mu$ < 20 GeV/$c$, respectively.

### Signal definition

The CCE, $\bar{\nu}H \to \mu^+ n$, signal is a subset of the quasi-elastic-like (QELike) sample in antineutrino scattering events. Precisely, 'QELike' events are those with a $\mu^+$ and only nucleons and remnant nuclei in the final state. In addition, events with energetic protons that produce reconstructable tracks, corresponding to a kinetic energy threshold of about 120 MeV, are excluded from the 'QELike' category in the antineutrino sample. The remaining charged-current antineutrino events, with other final-state particles or with sufficiently energetic protons, are labelled as 'non-QELike'. The QELike category contains, apart from the signal events, CCQE interactions on protons bound in carbon nuclei, interactions in which mesons are produced but later absorbed in the nucleus, and multinucleon knockout events (2p2h), in which two target nucleons participate in the scattering process. The antineutrino QELike category is naturally neutron-rich because of the net negative charge transferred to the target. Neutron detection is inefficient and does not give a strong constraint on neutron energy; therefore, it is not possible to exclude classes of events with multiple neutrons by counting detectable neutrons or measuring their energies. Signal events are instead constrained by the consistency of the direction of the neutron with the kinematics of the charged-current elastic scattering signal process.

### Reference interaction and detector model

A complete Monte Carlo simulation of the MINERvA detector, data acquisition and event reconstruction chain was developed to realistically predict the detector efficiency and resolution. The detector was simulated with the GEANT4 toolkit[41]. Signal and background events are generated by the GENIE[42] v.2.12.6 neutrino event generator with data-driven modifications. GENIE describes the carbon nucleus with a relativistic Fermi gas (RFG) model that assumes the nucleons are free, with the Fermi momentum maximum of $K_F = 221$ MeV. The single nucleon momentum is extended beyond the Fermi momentum cutoff to account for short-range correlations[62]. For the CCQE reactions, the RFG model is reweighted to the prediction of a spectral function model[50,63] used in NuWro[49], which incorporates both the shell structure of the nucleus and the nucleon correlation. Without the spectral function features, the initial state model is not sufficiently consistent with the carbon CCQE background to provide an accurate constraint. The reference CCE and CCQE cross-sections are modelled with a dipole axial form factor with $M_A = 0.99$ GeV/$c^2$ and the BBBA2005 (ref. [31]) vector form factors. The Valencia 2p2h model[44] used to describe the knockout of multiple nucleons is tuned by adding a large enhancement to match other MINERvA measurements[43]. The FSI is simulated by the GENIE INTRANUKE-hA[42] model. The model emulates a full-scale intranucleus cascade[64] in a way that is easily varied for systematic uncertainty studies. Some of the FSI reactions that change the final-state particle content in the detector are the inelastic FSI, pion production, pion absorption and pion and nucleon charge exchange. The elastic component of the FSI simulation in the CCQE channel is replaced with the no-FSI component through reweighting to circumvent an error in the INTRANUKE-hA kinematic implementation[65]. Non-CCQE interactions, such as resonant pion production, are based on the Rein–Seghal[66] model, and the deep inelastic scattering is based on the Bodek–Yang[67] model.

### Neutron interaction and reconstruction

Neutrons produced in neutrino interactions can interact in the detector through elastic or inelastic interactions on hydrogen or carbon. Neutron capture rates are negligible at MeV energies[68]. Elastic scattering on hydrogen $n^1\text{H} \to np$ or inelastic scattering on carbon, such as $n^{12}\text{C} \to np^{11}\text{B}$, can produce protons with enough energy to be observed in the detector. Neutron interactions are modelled using GEANT4. However, GEANT4 predictions of some inelastic neutron-C cross-sections are not in agreement with measurements of neutron scattering on scintillators at energies similar to our signal reaction[69]; therefore, we reweight the probability of neutron scattering from the GEANT4 prediction to be consistent with the recent models[70] that do agree with these data.

The MINERvA experiment has developed algorithms to reconstruct energy deposits from neutrons that produce ionizing particles by scattering off the material in the detector[37]. The daughter particles, primarily protons, are usually observed as isolated hits in the detector, far from the interaction vertex. If the particle is energetic enough to pass through two or more planes, a three-dimensional position of the interaction can be reconstructed and the direction from the vertex measured. The angle between the reconstructed and true neutron directions was assessed with the detector simulation. The distribution can be modelled as the sum of two exponential distributions $N(\theta) = A_1/\Theta_1 \exp(-\theta/\Theta_1) + A_2/\Theta_2 \exp(-\theta/\Theta_2)$, where the values of $A_1$ and $A_2$ are such that 40% of the neutrons have a narrow $\Theta_1 = 6°$ exponential slope and the remainder have a $\Theta_2 = 20°$ exponential slope. The heavy tail of this distribution probably results from the undetected scatters of neutrons in the detector before a detected scatter is observed. The intrinsic neutron angular resolution is convoluted with physics-driven neutron direction to enable constraint on the carbon CCQE events.

Extended Data Fig. 2 illustrates the schematics of the angular variables $\delta\theta_R$ and $\delta\theta_P$, and shows raw data and fitted model distributions.

The $\delta\theta_P$ distribution is symmetric about $\delta\theta_P = 0$, largely due to Fermi momentum in the CCQE background, whereas for the CCE signal the smearing is dominated by secondary interaction effects. There are bulk shifts in the peak positions of $\delta\theta_R$ distribution from the non-CCQE components due to the neutron assumption required in the direction prediction. Whereas the transverse component of the predicted neutron momentum is fixed by the muon's transverse momentum (equation (3)), the component parallel to the neutrino beam depends on the invariant hadron mass of the assumed final state ($M_n^2$ in equation (2)), resulting in a systematic shift when the final-state hadronic system is not a single neutron. Extended Data Fig. 3 shows the Monte Carlo event rates for a few interaction types at different $Q^2$.

## Event selection

Events with a muon track starting in the scintillator tracker region and ending in the MINOS ND with a positive charge identification are selected for analysis. Events with additional tracks formed at the vertex are rejected because they indicate the presence of charged mesons or energetic protons. The recoil energy is defined as the total energy outside a 100 mm sphere from the vertex and not associated with the muon track. A $Q^2$-dependent maximum recoil energy cut was applied to reduce the fraction of non-QELike background (Extended Data Table 1).

As described above, neutrons become detectable when they inelastically interact in the detector to create secondary particles. Neutron candidates in the detector are reconstructed by an algorithm that combines detector hits not attached to an interaction vertex and known particle tracks. Neutron candidates whose hits span two views or more produce three-dimensional position information, and only events with such candidates are selected for this analysis. Even at this early stage, the sample is already dominated by events with a final-state neutron from an antineutrino interaction.

In the CCE $\bar{\nu}$H reaction, the energy transfer $q_0$ from the neutrino to the initial state proton is proportional to the $Q^2$:

$$q_0 = \frac{Q^2}{2M_n}. \tag{9}$$

Leading neutron candidates with detectable energy above this physics limit cannot originate from the signal reaction, so such events are rejected. The QELike selection also restricts the number of isolated energy deposits outside the vertex region. Events with a single deposit whose energy resembles proton energy deposits with at least 10 MeV per strip are defined to be 'QELike'. Because of the inefficient neutron selection, events with multiple energy deposits are predominately events with particles other than neutrons, such as photons from $\pi^0$ and mis-reconstructed $\pi^\pm$. Events with multiple candidates are defined to be in the 'non-QELike' sideband sample.

Control regions are formulated on the basis of opening angular separations between the reconstructed neutron candidate and the expected neutron direction under the hydrogen hypothesis. The angular variables are defined according to equation (7), with regions formulated with varying fractions of QELike backgrounds, as shown in Fig. 1. The signal selection occupies the central $-10° < \delta\theta_P/\delta\theta_R < 10°$ region, whereas the 'non-QE and mesons' region encompasses all angular space outside the named regions. A bin-by-bin background constraint is performed for both the QELike and non-QELike events in bins of $Q^2_{QE}$, which is the $Q^2$ computed under the hydrogen hypothesis. Equation (10) shows the $\chi^2$ function minimized in the fit:

$$\chi^2 = \sum_{S,i} \frac{\left(\left[\sum_C w_{C,i} N^{mc}_{C,S,i}\right] - N^{data}_{S,i}\right)^2}{N^{data}_{S,i}} + \lambda_S \sum_C \sum_{j=1}^{N-2} (w_{C,j} + w_{C,j+2} - 2w_{C,j+1})^2. \tag{10}$$

where $N^{mc}_{C,S,i}$ is the Monte Carlo event rate in the $i$th $Q^2_{QE}$ bin in the interaction type category $C$, and in the angular control sample $S$. $N^{data}_{S,i}$ is the data rate in the $i$th $Q^2_{QE}$ bin in a control sample $S$. Each category used in the fit receives a weight $w_{C,i}$; categories not fitted receive a constant weight of 1. A regularization term is added to the $\chi^2$ function to ensure the weights across the bins change smoothly. $\lambda_S$ is the regularization strength and is a meta parameter tuned to ensure consistency between the fit and validation regions. To do so, the fit is performed on the central value model for a range of $\lambda_S$. An average $\chi^2$ for the validation regions, weighted by their event rates, is then calculated. The $\chi^2$ from the QE validation region dominates this calculation due to its high statistics. The value of $\lambda_S$ is chosen through an L-curve study. A separate fit using these parameters is done for every source of systematic uncertainty to evaluate its effect on the fit.

The QELike categories constrained are CCQE on carbon, 2p2h and resonant events whose pions are absorbed through FSI. The CCE signal category is not tuned because its contribution is small in the control regions. QE carbon is the primary background type in the QELike sample. Events with mesons in the final states, such as single $\pi^0$, $\pi^\pm$ and multiple pions, are primarily constrained with the non-QELike sample. The control regions used for fitting are the 'QE', 'non-QE' and 'non-QE and mesons', in the QELike sample, and all regions combined in the non-QELike sample. The QE and non-QE validation regions in the QELike sample are used to assess the goodness-of-fit. The results of the fit and the ratio to the fitted models in the validation regions are shown in Extended Data Fig. 4. The total event rate in the non-QE validation region is about one-fifth of the QE validation region.

## Validation of CCQE model and background subtraction technique using $\nu_\mu$ events

The validity of the background constraint techniques with the above model was tested using a neutrino CCQE sample with no CCE scattering from the free hydrogen. Events with a negatively charged muon matched to MINOS and a reconstructed proton track are selected. As with the CCE reconstruction, we compute the theoretical nucleon direction, this time a final-state proton, on an event-by-event basis and compare it with the proton track direction to classify the event according to the same angular region scheme. The track reconstruction efficiency at MINERvA for protons scales with their momenta, and below 450 MeV/$c$ (100 MeV in kinetic energy) the protons are no longer reconstructed. Therefore, QE protons with $Q^2 < 0.2$ (GeV/$c$)$^2$ are very rare, with events dominated by 2p2h background events whose reconstructed $Q^2$ based on the QE hypothesis are small.

A fit was performed using the control samples. Extended Data Fig. 5 shows both the fitted neutrino event ratio to the simulation in the signal and QE regions analogous to Fig. 2 and the $\delta\theta_R$ in this sample. The fitted Monte Carlo adequately describes data when proton tracks from CCQE are reliably reconstructed. The small discrepancy at $Q^2_{QE}$ between 0.2 (GeV/$c$)$^2$ and 0.4 (GeV/$c$)$^2$ is covered by a 100% uncertainty in the 2p2h component in that restricted region, as shown by the $\delta\theta_R$ plot at the bottom of the figure, and is the probable source of the discrepancy, given that the 2p2h component only begins to contribute significantly in this $Q^2_{QE}$ range. Such a systematic deviation in the carbon model would be well within the systematic uncertainties applied to our CCE background analysis in the antineutrino beam.

## Unfolding

The signal event rate obtained after background subtraction contains smearing effects from detector resolution. This analysis, similar to many recent results from the MINERvA Collaboration, undoes the smearing by means of an iterative unfolding algorithm developed by D'Agostini[52]. The inputs to the algorithm are the smearing matrix obtained through the Monte Carlo study of the true and reconstructed $Q^2$ in the signal sample and the number of iteration steps, $N_i$, as a regularization parameter. When $N_i$ is too small, the unfolded distribution

will not have converged to the true kinematic distribution, but higher values of $N_i$ increase statistical variances of the reconstructed distribution. To find the value of $N_i$ that balances model bias and statistical uncertainties, we form statistical universes of a toy model obtained by reweighting the Monte Carlo signal sample. The toy model differs from the CCE model by about 10% at lower $Q^2$, and 20% at higher $Q^2$, to mimic the shape of the discrepancy between data and the model. An ensemble of pseudo-experiments is created by changing the value in each bin of the toy model to a random number generated by a Poisson distribution, with the mean set at the central value of the toy model in that bin. A total of 1,000 statistical universes have been made and $N_i = 4$ minimizes the median and mean of the $\chi^2$ of the result of the pseudo-experiments compared to the hypothesis of the toy model.

The unfolded data are corrected for the predicted efficiency obtained from the Monte Carlo study. The efficiency, shown in Extended Data Fig. 6, increases with higher $Q^2$ until 0.5 (GeV/$c$)$^2$ as the rate of inelastic interactions that produce protons over the reconstruction threshold increases. For $Q^2$ greater than 0.5 (GeV/$c$)$^2$, however, the signal efficiency drops due to two factors. First, the neutron angle with respect to the neutrino direction increases, and therefore the neutron passes through less detector material on average before exiting the detector. Second, the muon is produced at a wider angle with respect to the neutrino and therefore is less likely to be reconstructed in the downstream MINOS ND.

## Uncertainties

Extended Data Fig. 7 shows the breakdown of the fractional systematic and statistical uncertainties in our measurement. We assess systematic uncertainties in six broad categories. Modelling uncertainties in GENIE are broken into the FSI and cross-section model uncertainties. The muon reconstruction uncertainties account for resolutions in the muon energy, direction and the reconstruction efficiency. The low recoil fit category is due to uncertainties in the tuned 2p2h model. The 'other' category includes target mass uncertainties and a set of particle response uncertainties. The largest particle response uncertainties are related to the neutron interactions in the detector. The 'neutron reweight' uncertainty is due to the neutron elastic and inelastic cross-sections assumed in the detector; it is assessed for each neutron and assigned a larger value for lower kinetic energy. The neutron interaction uncertainty accounts for the discrepancy in the energy deposited in the neutron candidates we observed in a previous study[37]. These are the largest uncertainties in the 'other' category, and they mainly affect the low $Q^2$ cross-section. This analysis is dominated by the statistical uncertainty, in part because of the large background subtraction.

## Cross-section calculation and the $z$ expansion fit of $F_A(Q^2)$

The cross-section measured in this analysis is the flux-integrated cross-section for the geometry of the MINERvA detector. The measured event and predicted background rates are shown in Supplementary Table 1. The efficiency corrected event rates are divided by the total number of hydrogen atoms and protons on target (Supplementary Table 2). The measured cross-section is shown in tabulated form in Supplementary Table 3, and the total covariance and statistical-only covariance matrices of the measurement are shown in Supplementary Tables 6 and 7, respectively.

Calculating the integrated cross-section in each bin needs to account for the muon phase space cuts described above. The effect of the phase space restriction is manifested as a restricted range of neutrino energy available in the selected sample at each $Q^2$ point, ultimately reducing the differential cross-section at each $Q^2$ bin because the acceptance-corrected event rate is divided by the fully integrated neutrino flux for this measurement. Extended Data Fig. 8 (right) illustrates the accepted neutrino energy at each $Q^2$. Therefore, obtaining a fit of $F_A$ requires the convolution of equation (4) with the antineutrino flux from the RHC configuration of the NuMI[34] neutrino beam with

the $Q^2$-dependent energy cutoff. The vector form factors used in this analysis are parameterized from electron scattering data[31] used commonly by neutrino Monte Carlo generators[42,49,58].

The axial vector form factor is also fitted using the $z$ expansion[59] formalism. We adopt the procedure of a fit to deuterium data by Meyer et al.[24]. The $z$ expansion for $F_A$ is a polynomial of $z$ with coefficients $a_k$ reproduced here for convenience.

$$F_A(Q^2) = \sum_{k=0}^{k_{max}} a_k z^k \tag{11}$$

$$z = \frac{\sqrt{t_{cut} + Q^2} - \sqrt{t_{cut} - t_0}}{\sqrt{t_{cut} + Q^2} + \sqrt{t_{cut} - t_0}} \tag{12}$$

$$\sum_{k=n}^{\infty} k(k-1)\ldots(k-n+1)a_k = 0, \, n \in (0, 1, 2, 3) \tag{13}$$

Values of the coefficients $a_k$ are constrained by $F_A(0) = -1.2723$, and a set of four coefficient sum rules (equation (13)) determined by the high $Q^2$ behaviour required by QCD[71,72]. For a given value of $k_{max}$, $N_a = k_{max} - 4$ coefficients are free parameters: $(a_1, \ldots, a_{k_{max}-4})$. The error function involved in the fit is a $\chi^2$ function with a regularization term of the form $\lambda \left[ \sum_{k=1}^{5} \left( \frac{a_k}{5a_0} \right)^2 + \sum_{k=5}^{k_{max}} \left( \frac{ka_k}{25a_0} \right)^2 \right]$ based on the expected fall off of $a_k \propto k^4$ at large $k$ (ref. [24]). We choose $t_0 = -0.75$ (GeV/$c$)$^2$ in our analysis to centre our data points around $z = 0$. Small variations in $t_0$ have no effect on the final fit results. $k_{max}$ and $\lambda$ are parameters that affect the functional form of $F_A$ and the uncertainties of the form factor. Larger $k_{max}$ increases allow for a more 'curvy' $F_A$ that might overfit and follow statistical variations of the data, whereas a non-zero $\lambda$ suppresses this behaviour for larger $k_{max}$. At $k_{max} = 6$, the value of $\lambda$ has little effect on the low $Q^2$ behaviour of the fit but prevents $F_A$ from adequately describing the data at larger $Q^2$. The fit at $k_{max} = 8$ enables higher $Q^2$ fit to follow the data points, but also exposes the low-$Q^2$ region to a small dependence on $\lambda$, which manifests as a few per cent variation in the calculation of nucleon axial radius measured on a proton, sufficiently covered by the uncertainties from the fit. The behaviour of the regularization term and the data $\chi^2$ term were studied and the fit at $\lambda = 0.13$ was chosen according to the maximum curvature, or L-curve, criterion[73]. Extended Data Fig. 8 (left) shows the final fit to data. In general, varying the $z$ expansion parameters, such as $k_{max}, t_0$ and $\lambda$, results in changes below 10% of the total uncertainty. Supplementary Tables 4 and 5 show the fit result and the correlation matrix using $k_{max} = 8$, $\lambda = 0.13$ and another fit with $k_{max} = 6, \lambda = 0$.

## Data availability

The extracted cross-section data reported in this study are available in the extended figures and tables section, and will be made available for public access on https://minerva.fnal.gov/data-release-page/.

## Code availability

The MINERvA Collaboration develops and maintains the code used for the simulation of the experimental apparatus and statistical analysis of the raw data used in this result. This code is shared among the collaboration, but not publicly distributed. Enquiries regarding the algorithms and methods used in this result may be directed to the corresponding author.

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

**Acknowledgements** All authors are members of the MINERvA Collaboration. This document was prepared by members of the MINERvA Collaboration using the resources of the Fermi National Accelerator Laboratory (Fermilab), a US Department of Energy, Office of Science, HEP User Facility. Fermilab is managed by Fermi Research Alliance, LLC (FRA), acting under contract no. DE-AC02-07CH11359. These resources included support for the MINERvA construction project, and support for construction also was granted by the United States National Science Foundation under award no. PHY-0619727 and by the University of Rochester. Support for participating scientists was provided by NSF and DOE (USA); by NSERC (Canada); by CAPES and CNPq (Brazil); by CoNaCyT (Mexico); by Proyecto Basal FB 0821, CONICYT PIA ACT1413, and Fondecyt 3170845 and 11130133 (Chile); by CONCYTEC (Consejo Nacional de Ciencia, Tecnología e Innovación Tecnológica), DGI-PUCP (Dirección de Gestión de la Investigación-Pontificia Universidad Católica del Peru) and VRI-UNI (Vice-Rectorate for Research of National University of Engineering) (Peru); NCN Opus grant no. 2016/21/B/ST2/01092 (Poland); by Science and Technology Facilities Council (UK); by EU Horizon 2020 Marie Skłodowska-Curie Action. We thank the MINOS Collaboration for use of its near detector data. Finally, we thank the staff of Fermilab for support of the beamline, the detector and computing infrastructure.

**Author contributions** The operation, Monte Carlo simulation and data analysis of the MINERvA Experiment are carried out by the MINERvA Collaboration with contributions from all collaborators listed as authors on this manuscript. The scientific results presented here have been presented to and discussed by the full collaboration, and all authors have approved the final version of the manuscript. T.C. and A.B. first proposed this analysis to the authors. D.R. developed the analysis tools with input from T.C. and A.O. T.C. performed the analysis of data and simulation described in this work. T.C. and M.L.M. performed the fits to the data. T.C. and K.S.M. wrote the manuscript.

**Competing interests** The authors declare no competing interests.

**Additional information**
**Correspondence and requests for materials** should be addressed to T. Cai or K. S. McFarland.

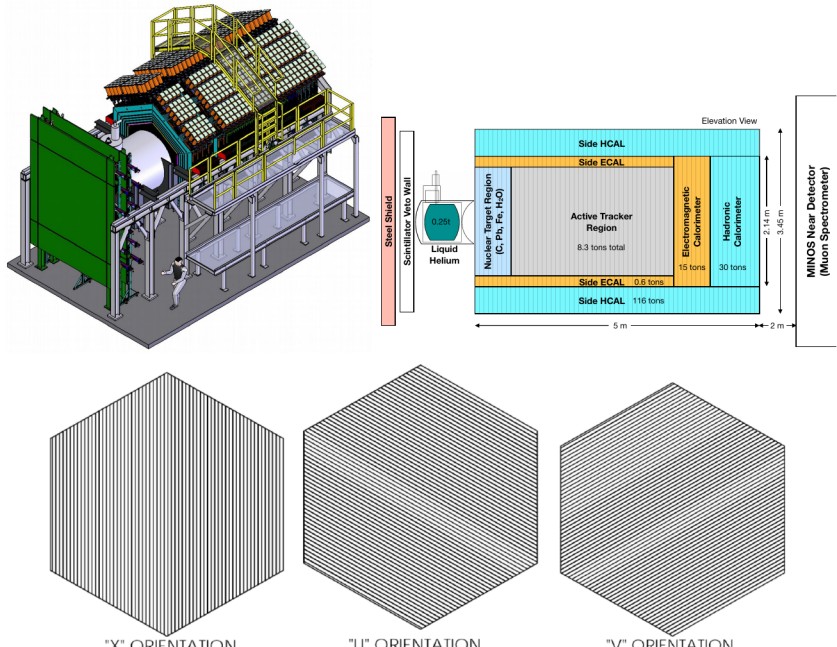

**Extended Data Fig. 1 | Illustration of MINERvA detector.** A 3D illustration of the MINERvA detector (left), a flat cross-sectional portrayal (reproduced from ref.[11], right). The detector comprises 120 hexagonal modules, each consists of an Inner Detector (ID), side Electromagnetic Calorimeter (ECAL), and side Hadronic Calorimeter (HCAL) arranged outwards from the center. Horizontally along the beam direction in the ID region are four sub-detectors: nuclear target detector, active scintillator detector, the ECAL and the HCAL. Each scintillator plane in the detector is arranged in one of "X", "U", or "V" (bottom) orientations. A liquid helium tank is located upstream of the nuclear target detector. Downstream of the MINERvA detector is the magnetized MINOS[36] near detector acting as MINERvA's muon spectrometer.

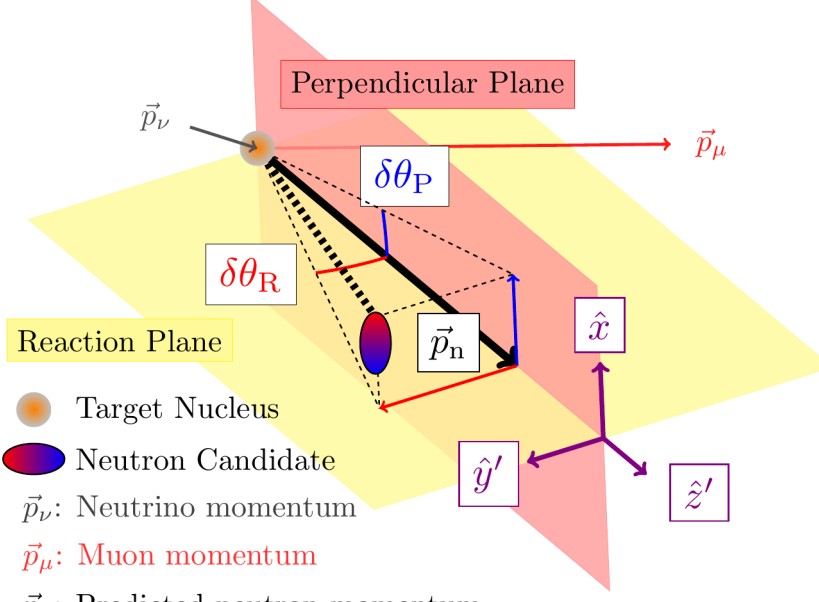

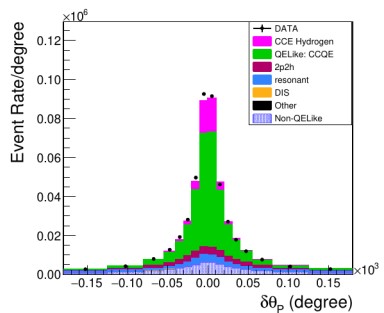

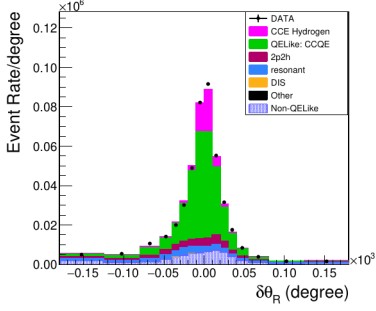

**Extended Data Fig. 2 | Definition of $\delta\theta_P$ and $\delta\theta_R$, and the event distributions.** (top) Angular variable definitions. The variables $\delta\theta_R$ and $\delta\theta_P$ are defined according to the rotated reference frame (($\hat{x}, \hat{y}', \hat{z}'$)). Distributions of

(bottom left) $\delta\theta_P$ and (bottom right) $\delta\theta_R$ in the QELike sample normalized to bin width, after sideband fits. The vertical error bars around the data points represent 1 standard deviation due to statistical uncertainty.

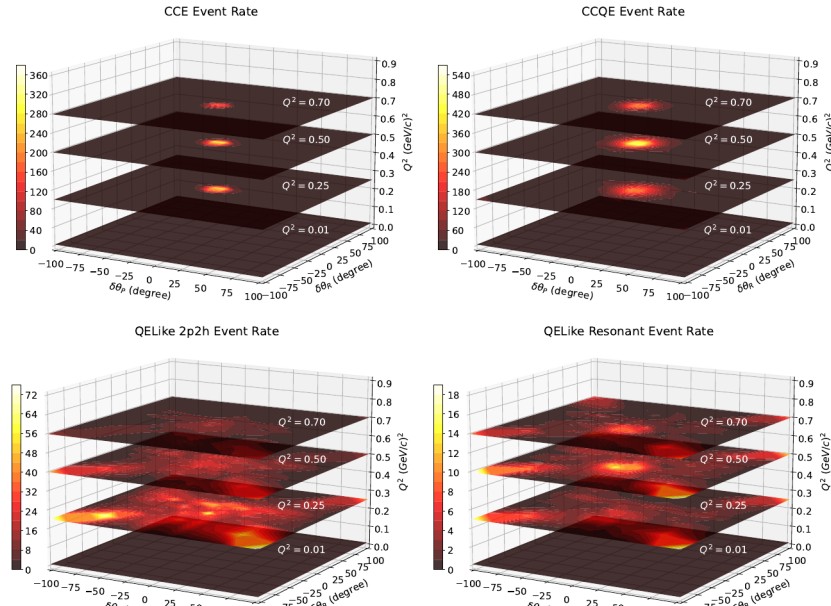

**Extended Data Fig. 3 | Simulated event rate in $\delta\theta_P$ – $\delta\theta_R$ plane in selected $Q^2$ analysis bins.** Heat map showing Monte Carlo event rate for CCE, CCQE, QELike 2p2h, and QELike Resonant interaction models in the $\delta\theta_R$-$\delta\theta_P$ plane at a few slices of $Q^2$. CCE events are concentrated around origin, while the CCQE events have broader spread. Both QELike 2p2h and resonant events show diffused structure going out to larger $\delta\theta_R$ and $\delta\theta_P$ regions. The color scales in the heat maps are different because of different event rates for each subsample.

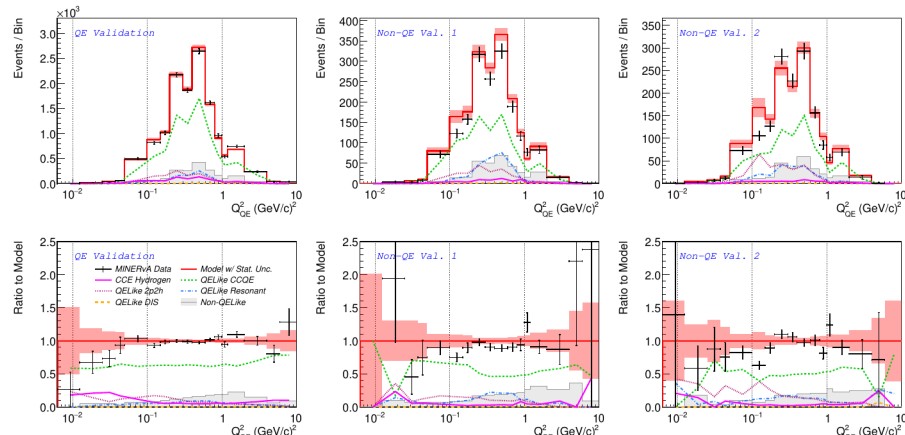

**Extended Data Fig. 4 | Post-fit event rates and ratios in the validation regions.** The "Non-QE validation" region is shown separated into two sub-regions. "Non-QE Val. 1" spans $|\delta\theta_P| < 20°$, "Non-QE Val. 2" occupies $20° < |\delta\theta_P| < 55°$. The vertical error bars around the data points and the error band around the model prediction account for 1 standard deviation due to statistical uncertainty. The CCE signal and the regions used in the background fits are shown in Figs. 2 and 3, respectively.

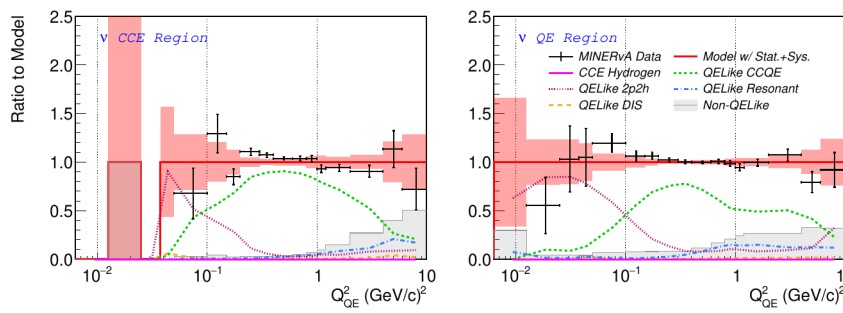

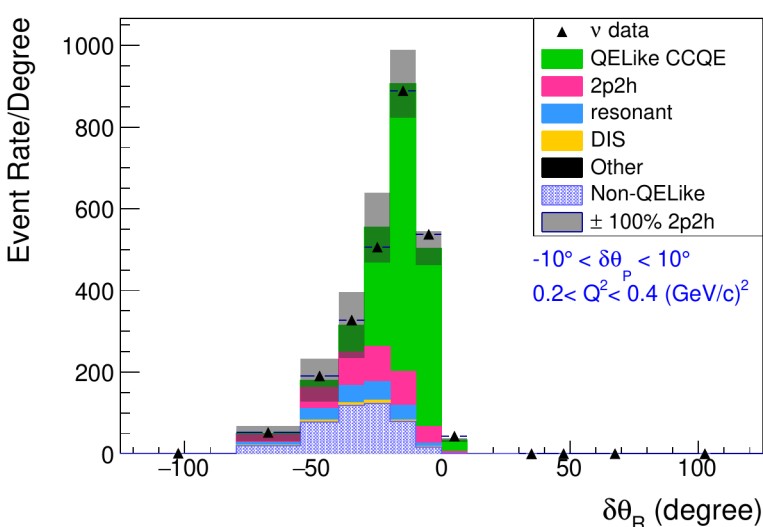

**Extended Data Fig. 5 | Background constraint method tested on a neutrino sample.** (Top left) Signal and (top right) QE region distributions of the neutrino sample that looks for proton final states after fit. The vertical error bars around the data points and the error band around the model prediction account for 1 standard deviation due to statistical and systematic uncertainties. (Bottom) Application of $\pm 100\%$ shift in 2p2h (gray band) on $\delta\theta_R$ for events in a $-10° < \delta\theta_P < 10°$ and 0.2 $(GeV/c)^2 < Q^2 < 0.4$ $(GeV/c)^2$ slice. The analog to the CCE selection selects between $-10°$ and $10°$ in $\delta\theta_R$.

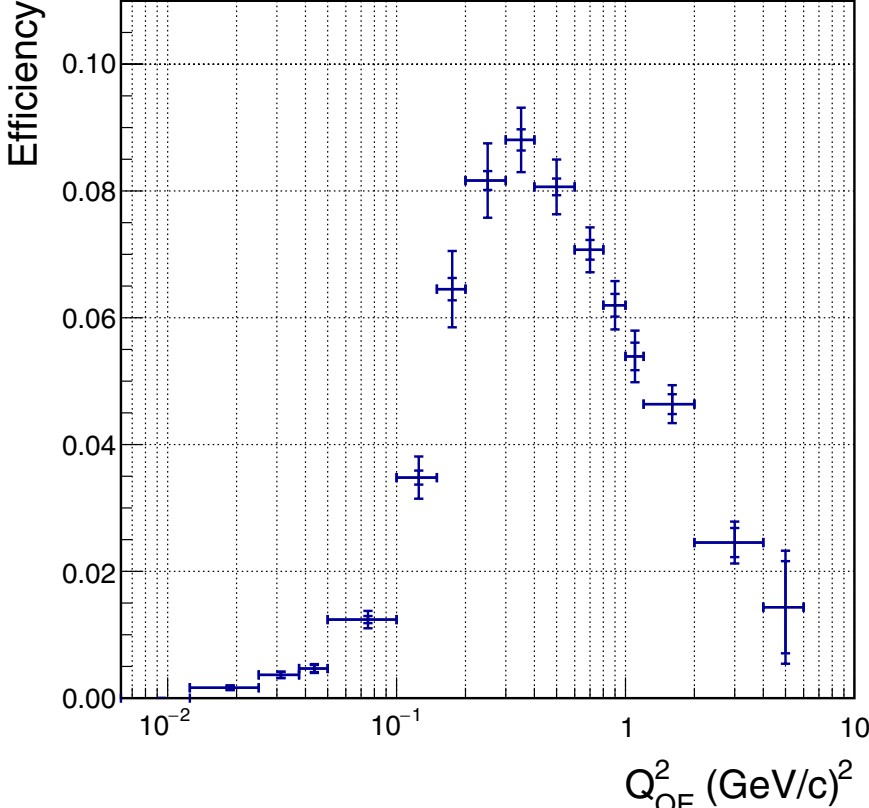

**Extended Data Fig. 6 | Signal event selection efficiency.** CCE signal efficiency as a function of $Q^2$. The inner error bar on each data point accounts for the statistical effect of a Poisson standard deviation, while the full error bar account for all sources of systematic uncertainties.

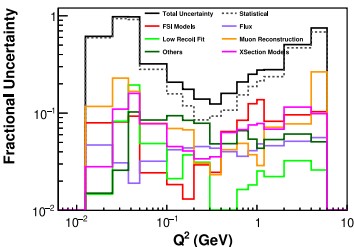
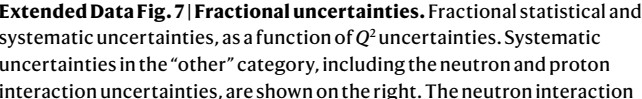
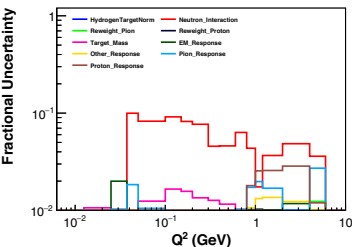

**Extended Data Fig. 7 | Fractional uncertainties.** Fractional statistical and systematic uncertainties, as a function of $Q^2$ uncertainties. Systematic uncertainties in the "other" category, including the neutron and proton interaction uncertainties, are shown on the right. The neutron interaction systematic accounts for the neutron secondary interaction uncertainties in detector. The leading interaction channels, such as $(nC, Bnp)$, $(nC, 3\alpha)$, and $(nC, n'C\gamma)$, are assigned 10% to 15% uncertainties below a kinetic energy of 100 MeV.

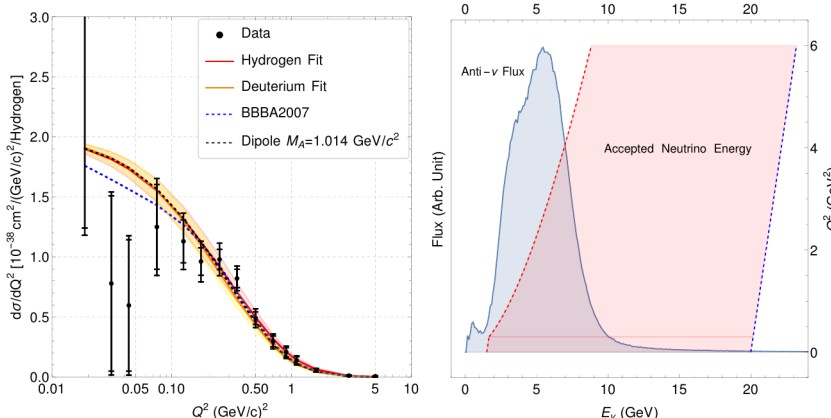

**Extended Data Fig. 8 | Cross section and the $Q^2$-dependent flux cut.** (left) Measured cross-section and theory predictions, (right) Regions of neutrino energy and flux in signal selection at each $Q^2$. The inner error bars on the data points account for 1 standard deviation due to statistical uncertainty only, and the full error bars include contribution from all sources of systematic uncertainties.

**Extended Data Table 1 | Recoil energy cut**

| Condition | $E_{\mathrm{max}}^{\mathrm{recoil}}$ (GeV) |
|---|---|
| $Q_{\mathrm{QE}}^2 < 0.3\ (\mathrm{GeV}/c)^2$ | $0.04 + 0.43 Q_{\mathrm{QE}}^2 / (\mathrm{GeV}/c)^2$ |
| $Q_{\mathrm{QE}}^2 < 1.4\ (\mathrm{GeV}/c)^2$ | $0.08 + 0.3 Q_{\mathrm{QE}}^2 / (\mathrm{GeV}/c)^2$ |
| $Q_{\mathrm{QE}}^2 > 1.4\ (\mathrm{GeV}/c)^2$ | $0.50$ |

The recoil energy ($E^{\mathrm{recoil}}$) is the total energy outside a 100 mm sphere from the vertex and not associated with the muon track. The $Q^2$-dependent maximum recoil energy ($E_{\mathrm{max}}^{\mathrm{recoil}}$) cut reduces the fraction of non-QELike background.