## [Peer Review File · Nature]

Manuscript Title: Measurement of the axial vector form factor from antineutrino-proton scattering

Reviewer Comments & Author Rebuttals

Reviewer Reports on the Initial Version:

Referees' comments:

Referee #1 (Remarks to the Author):

The paper, "First measurement of the nucleon axial vector form factor in antineutrino scattering from hydrogen" by the Minerva collaboration, discusses a measurement of the $\bar{\nu}_p$ to μ^+ cross section from hydrogen. This paper discusses two main points: First is the extraction of the axial form factor $F_A(Q^2)$ and second, a determination of the mean of the axial charge radius squared.

1. In the "Criteria for publication", Nature states, "In general, to be acceptable, a paper should represent an advance in understanding likely to influence thinking in the field. . ." This is a very high mark. The abstract tells me why the transitional axial form factor is being measured this way and tells me that this "is an important input to current and future neutrino oscillation experiments". As only an input to future experiments, I do not see the manuscript reaching this high mark. The first three paragraphs of Sec. 1 go on to explain why this is an important input to other results. The second paragraph concludes, " $F_A(Q^2)$ is key input to the models that strongly affects the rate of charged-current scattering from nucleons." This needs to be explained more to quantify why this measurement is likely to influence thinking in the field. These are valuable results and should be published elsewhere. I also don't rule out that a better presentation of the importance of these results might be possible, and I could reconsider, but I do not see it now.

2. Does the manuscript have flaws which should prohibit its publication?

There are no serious flaws with the analysis presented in the manuscript that should prevent its publication. In fact, I am strongly in favor of its publication in any number of other journals. There were some places where more explanation would be helpful for the readers, and I will comment on some to them below.

3. For the determination of the axial form factor

a. The fit to extract the CCE signal could be better explained. In Figs. 2 and 3, my understanding is that

i. A separate fit is done for each bin in Q^2 .

ii. The fit is simultaneously made to three separate regions from Fig. 1. The three regions are QE, non-QE, and non-QE plus mesons.

iii. The fit uses Monte Carlo distributions of CCQE, Resonant, $2p2h$, DIS, non-QELike, and CCE events.

iv. The output of the fit gives the relative normalizations of the above components. These fits are

shown in Fig. 3.

v. These are then cross checked with the two validation regions to determine how good the fit really describes data in an unfitted region (not shown).

vi. Then, with the validated fit, the CCE signal is extracted from the signal region.

I know, however, that this reasoning is wrong in step (i), since only one χ^2 is quoted rather than one for each Q^2 bin.

b. What are the systematic uncertainties from this fit and using the chosen regions? Did the fit take into consideration the Monte Carlo statistics or only the data statistics? (Clearly with enough computer time, the Monte Carlo statistics can be made arbitrarily large, and this question doesn't matter.)

c. The fits shown in Fig. 2 (left) and 3 (top) have a lot of "white space" in the vertical scale that leaves me squinting to see the content.

d. Is "model" in Figs. 2 and 3 referring to simply the sum of the components CCQE, Resonant, 2p2h, DIS, non-QELike, and CCE events?

e. Since the entire argument of being able to identify event types in this paper depends on the event types having substantially different distributions in the $\delta\theta^R$ vs $\delta\theta^P$ plane, it would be nice to see what these distributions are for some bin in Q^2 (from Monte Carlo of course).

f. In Sec. 8, the modeling of the distribution in terms of one angular parameter with the $N(\theta)$ equation seems inconsistent with the discussion of $\delta\theta^R$ and $\delta\theta^P$ each being significant, unless it is meant to say that the A_1 term corresponds to $\delta\theta^P$ and A_2 corresponds to an additional widening from $\delta\theta^R$, but then there doesn't appear to be symmetry in $\delta\theta^R$, which this would imply.

g. How is Fig. 6 bottom normalized? Given the abundant statistics, the histograms many standard deviations from the data.

4. The discussion of the axial charge radius brings up a few questions:

a. It needs to be noted that $\langle r_A^2 \rangle$ is a $\lim_{Q^2 \rightarrow 0} dF_A/dQ^2$ —the limit is important—so the resolution on Q^2 and how low in Q^2 are both important. The former is available from the tables at the end of the document (although I would have liked to see it mentioned earlier). The latter, I didn't find.

b. In the discussion of the z-expansion formalism and the corresponding fit, why is $t_0 = -0.75$ (GeV/c)²? Hill and Paz, Phys. Rev. D 82, 113005 (2010) state that this is a free parameter, but also suggest that $t_0^{\text{opt}} = t_{\text{cut}} (1 - \sqrt{1 + Q_{\text{max}}^2/t_{\text{cut}}})$ which would be around -0.78 for $Q_{\text{max}}^2 = 5.0$ GeV². The Methods Section says that t_0 was chosen "to center our precise data points around $z = 0$." Which points are considered "precise"? From Tab. 4, these seem to be the points at larger Q^2 . Does this matter, as lower Q^2 points are closer to the $\lim_{Q^2 \rightarrow 0}$. Does this choice matter?

c. How was k_{max} chosen and was there a variation with k_{max} ? Which QCD sum rules determine 4 parameters (perhaps this should be obvious to me)? What happens in the case of $k_{\text{max}} = 4$, i.e, no fit—does the expansion reasonably agree with data?

d. Why are there bounds on the values of a_k ? In particular, the fit seems to converge, according to Tab. 2, with the a_k/a_0 not near the bounds, so are they necessary?

5. If a_0 is taken from $F_A(0) = -1.2723 \pm ?$ Is the uncertainty on this quantity meaningful in the fit? Would the Minerva data otherwise constrain $F_A(0)$?

6. Sec. 1, Theory, “In the limit of low negative four-momentum transfer squared, Q^2 , form factors are the Fourier transform of charge distributions.” This statement is viewed as incorrect by some theorists. See, for example, G. Miller, Phys. Rev. C 99 (2019) 3, 035202. It is a good motivational tool but needs to be acknowledged as a such.

7. Figure 6-top is critical to understand the discussion of any of the $\delta\theta_R$, $\delta\theta_P$ binning, therefore the authors should consider moving the figure to just before the discussion of Fig. 1. What is the angular resolution on the reconstructed muon track? This tells me how well the calculated neutron direction is determined.

8. Is Fig. 1 contains a lot of information. The caption should say that the pie charts are based on the Monte Carlo fit to the data (if this is the case). The other question is how many events are in each bin in this figure?

9. Sec. 5, p. 20, The energy distribution is as important as the central value. Knowing that it peaks at 5.5 GeV is nice, but not sufficient to calculate Q^2 . Alternatively, what is your resolution in Q^2 ?

10. Are the first two panels of Fig. 7 the same as are shown in Figs. 2 and 3?

Copy editing and clarity comments (no need to respond to these, just trying to be helpful):

11. P. 5, line after Eq. 1 is not a new paragraph.

12. P. 5, is it $\langle r^2 \rangle_A$ or $\langle r_A^2 \rangle$? I would say that it is the average of the (axial radius squared) and thus the latter formulation

13. The variable Q_{QE}^2 is first introduced in the labeling of Figs. 2 and 3 in the main paper, but it is not defined in the text until Sec. 10 in the methods section, leaving the reader to speculate until then.

14. Sec. 3, p 6, last line: it would be clearer to say that the CCE signal is exclusively a subset of the QELike events?

15. Figure 5 in Sec. 2 is the first figure mentioned in the text—shouldn't that make it Fig. 1? I must go all the way back to the Sec. 5 in Methods to see the figure. Perhaps rather than referencing the figure, just say that a more detailed description is given in the Methods section.

16. Sec. 2, p. 6, last paragraph, add a comma before the word "which": "... low energy protons, which can ..."

17. Fig. 2, p. 8: Half of the space in this figure is white space. Changing the vertical scale from $(0.0-5.0) \times 10^4$ would let the reader see details better (e.g. statistical error bars). Also, perhaps a log scale would be better so that one could see the contents of bins with $Q^2 < 8 \times 10^{-2}$?

18. Page. 8, 2nd line, "... did not scatter undetectably before detection" the word "detect" in one of the two instances should be replaced; although, I understand the meaning.

19. Page 10, paragraph below Eq. 3: The comma should be on the same line as " $F_A(0) = -1.2723$ "

20. Page 10, it is stated, "The fit at $\lambda = 0.13$ is well-behaved at the low and highest Q^2 range." I assume that you mean that in addition to the ends, it is well behaved throughout the Q^2 range?

21. Page 10, "The proton axial radius from the fit to this result is $r_A \equiv \sqrt{\langle r_A^2 \rangle} \equiv 0.73(12)$ fm." is a one sentence paragraph.

22. Page. 23, sentence 2. Here "Scattered particles" refer to secondary particles scattered by the (quasi-) elastically scattered neutron. In much of the rest of the paper, "scatter" refers to the primary neutron-neutrino scatter.

23. P. 23, in the equation for $N(\theta)$, is $A_2 = 1 - A_1$? The use of a percentage for A_1 makes it appear so, but then why not make it explicit.

24. Sec. 11, p 26, paragraph 2: Do you mean analogous to Fig. 2, not 1?

25. Don't use section headings as simply a mechanism for a new paragraph.

Referee #2 (Remarks to the Author):

Dear authors,

the method presented in this paper to measure the axial proton form factor is original, effective and very promising for the future neutrino long-baseline experiments.

Congratulations for this brilliant analysis!

The data are very valuable and their reliability is assured by the long experience in running Minerva experiment since many years. The analysis is well presented.

The collaboration has large expertise in measuring cross-section of neutrino-nucleus scattering and such expertise (notably in use of statistics and treatment of uncertainties) is reliably applied in this new analysis.

Regarding the methodology and robustness of the results I have few comments below for which I would like to have your feedback. Please, do not hesitate to correct me in case of misunderstanding from my side.

1) You highlight quite strongly in the abstract the larger form factor at high Q^2 in your results with respect to previous deuterium measurement.

I find this excess pretty surprising since the deuterium corrections should impact mostly the low Q^2 region. Maybe this excess comes more from the improved functional shape (Z-expansion) than from your new method/data?

Moreover in your analysis, in this high Q^2 region, the CCE signal is much lower than the QE-like Carbon background (looking for instance at Fig.2). And even the non-QE-like background is almost as large as the CCE signal there.

As a consequence, I would like to investigate better if any residual bias is possible in the estimation of the backgrounds or elsewhere. The next few questions go in that direction.

2) One possible source of bias could be the neutron detection efficiency as a function of Q^2 in the signal.

2.a) It is pointed out indeed that in Ref.[35] a discrepancy has been observed between data and MC. I wonder if the mis-modeling of neutron secondary interactions in the detector could be dependent on the neutron energy and thus, indirectly, on the Q^2 . This would induce a wrong estimation of Q^2 -dependent efficiency correction from MC and thus, ultimately, a bias in the estimated cross-section vs Q^2 . How robust is the control of this issue?

2.b) Another possible source of bias could be the Q^2 dependent cut of E^{recoil} (Tab1). It is my understanding that with this cut you could also affect the signal where the E^{recoil} is not a real recoil but it is due to CCE neutrons which interact already in the 100mm sphere. How much CCE signal you are cutting with Tab1 cuts and how much this cut impacts the Q^2 -dependency of the signal efficiency? If it's sizable, how you ensure that there is no-bias induced by this MC-based correction?

2.c) another possible source of bias could be the Q^2 -dependence of the muon efficiency. In the regions where large Q^2 bins are used, the efficiency for muon reconstruction may change quite a lot inside the bin and the MC-based correction is effectively an average over the assumed signal Q^2 distribution. Therefore the assumed Q^2 shape of signal in MC (if different from reality) can induce a bias in the efficiency correction and thus in the results. Could you demonstrate this effect is small?

3) One possible source of bias could be the estimation of background. The Q^2 distribution is fit from data in control regions but, if I understand well, the extrapolation as a function of $d\Theta_R$ and

dThetaP (from the control region to the signal region) relies on MC simulation of neutron final state interactions and on neutron secondary interactions in the detector (and again Ref[35] pointed out a discrepancy in the simulation). Could you comment on how well FSI and SI systematics are under control? Which systematics do you apply to FSI and SI?

3a) I'm also worried of the patch used to fix INTRANUKE-hA: I would not expect the elastic component to have similar kinematics of no-FSI. Could you show them?

3b) Sec.8 After having reweighted the GEANT4 predictions: what are the remaining assumed uncertainties on neutron SI? (Would be nice to report this into the paper as well)

3c) You mention that you rely on simulation to characterize the neutron direction (with the sum of two exponential). Do you have any data control sample to confirm this?

3d) The validation with neutrino events in Sec.11 does not really close successfully (large excess of events in signal region at low Q²) and this looks worrisome to me. You mention that this could be due to large uncertainties in 2p2h but such excess is present only in the signal region (low dThetaP, low dThetaR) and not in the control region. It seems to point to a mis-modeling of the extrapolation from control to signal region which rely on neutron FSI or SI modeling (which could be present in the antineutrino events also).

Further minor clarification on the analysis

- how do you take into account the CCE component in the control regions?
- how large (probably small?) is the background from neutrino interactions in this antineutrino sample. What is his Q² dependence and how well is under control?
- how the cuts of Table1 have been optimized?
- is unfolding done before or after background subtraction? Is the optimization of N_i taking into account also possible mismodeling of QELike background?

Concerning the clarity of the presentation, I miss two important and clear explications

1) I would clearly explain the kinematics first: how do you compute Q² (put the formula), how do you predict the expected CCE neutron direction (formula)?

Explain clearly at the beginning that for CCE you have such prediction due to nucleon at rest and you use it to search the neutron in the expected direction, while the backgrounds will have different (smeared/bias) distribution around such prediction.

2) Mention earlier about the measurement is convoluted with the flux

3) I find this statement a bit too strong:

“This result is the first significant measurement of the axial-vector form factor free of nuclear corrections or other theoretical assumptions”

You still have nuclear effects in the background subtraction, in the flux estimation and in the systematics on signal due to SI. I wonder what would be a better formulation for the above sentence... Maybe just acknowledging such nuclear effects explicitly right before/after that sentence would be enough...

Another major comment is about the title: here you measure the proton form factor. While the title mention the ‘nucleon’ form factor.

Minor comments on the clarity of the text:

- Sec1: I found the following sentence not precise “past extraction have assumed FA to follow the

dipole form factor, which now known not to be true for the electromagnetic factors". I would suggest to rewrite along the lines below

Actually various attempt of using other models have been already done like in [10] but also in Eur. Phys. J. C 53, 349–354 (2008) , Phys.Rev.C 101 (2020) 2, 025501 or others.

Also, I think 'not true' is weird wording. The dipole has always been considered a first order approximation. You may want to say that better or alternative models are now available.

- The validation test with proton (instead of neutron) is mentioned ins ec.3 but without any conclusion. please, put in the sec3 already if there is any left excess of events (which would be a sign of not closure of the test). If the validation works perfectly you could report just 0+/-stat uncertainty, but it is interesting to know immediately for the reader what is the level of precision of such powerful test.

- Sec 4: please list the most important systematics uncertainties here (very briefly). Please state that the theory will need, not only to apply the muon acceptance cuts and the Enu cuts, but also to convolute the prediction to the Minerva flux (and possibly a reference to it)

- Sec.8 "linear dependency of the invariant mass". I found this part confusing: a formula of how you predict/define thetaR would help the reader to understand.

- Sec.9 "Events with a single deposit whose energy resembles proton energy deposits with at least 10 MeV per strip are defined to be QELike". How do you distinguish such proton events from the ones where the proton is produced by the scattering of a neutron from CCE?

- Fig.10. In my opinion the neutron SI systematics is (with FSI) the most crucial systematics for the reliability of the analysis. It also has large impact in the intermediate Q2 region. I think it deserves its own line in the left plot of Fig.10. (And as mentioned above, I strongly suggest to add explicitly what is the assumed uncertainty after GEANT4 corrections.)

- Sec2: the first figure mentioned in Fig.5, why you do not put that as Fig.1 ?

- Sec3: Fig.2 and fig3 are not mentioned anywhere in the text here (only much later)

- Sec4: typo with comma on the other line after $FA(0)=-1.2723$

- Sec.7: "Fermi momentum set to $K_F=221\text{MeV}$ " looks weird. Fermi momentum is a distribution, I gues you may want to say 'with maximum cutoff at $K_F=...$ '

- Sec.8: "the the" repeated twice

- Sec8: 'largely due to Fermi momentum' → "largely due to Fermi momentum for background, while for signal smearing is dominated by FSI/SI effects" Correct? I would write something like that, more explicit.

- Sec.10: "Others region" → "others region". Actually I'm not sure you defined it as 'others'. Osn't it

non-QE?

- Sec11: "450 MeV/c(100MeV)" you didn't say what is the number in parenthesis (I guess kinetic energy?). Please mention it

Fig.8: "region 1" is not defined, probably a left-over from internal terminology

References

I found that some of the choices of references is not obvious. Here is a list of examples/suggestions:

- Abstract:

[1] it is weird to put only this paper. It is maybe the most recent but it for proton only (while electromagnetic form factors for neutrons are different and in this sentence you mention both protons and neutrons). A reference explaining the theory of electromagnetic form factors would look more appropriate in this sentence.

[9,10] I found weird to cite [10] here where you discuss Deuterium corrections. I think [10] apply previous Deuterium corrections from [9], it does not develop new ones. Maybe I am wrong?

[11-13] [11] and [13] are review and they are fine but then [12] is just one single result of calculation and I guess there different results available. Why you chose that one?

[15-19] Here you include general papers on DUNE, HK, NOVA and JUNO. Why JUNO? Is F_A relevant for JUNO? Also for T2K you cite an actual oscillation analysis results. I would quote the general T2K experiment paper (as you do for the other experiments)

- Section1

[1] same comment as before

[20] seems to me unnecessary

[10] and [10,24] [24] is a general review but why you include only the example of [10]? There are other papers which extracted F_A from those data (and beyond) like Phys.Rev.C 101 (2020) 2, 025501 or Eur. Phys. J. C 53, 349–354 (2008)

[11-13] same comment as for the abstract

[29-31] also here I think there are other evaluations, I think. As in Phys.Rev.C 101 (2020) 2, 025501

-Section3

[41-46] a strange selection of references what is the logic of this specific selection?

[47] I would add also [61] here.

- Section7

[61], [47] I would add [48] here

- Section[8]

[39] doesn't look appropriate. Here actually we see quite relevant neutron capture for different nuclei than Carbon... It does not discuss Carbon I think.

Referee #3 (Remarks to the Author):

This is a very exciting experimental result. For decades it has been very difficult to extract the nucleon axial form factors without worrying about the nuclear effect due to the large nuclei used. MINERvA Collaboration's antineutrino scattering from hydrogen is really changing the game for what we can learn about nucleon axial form factors from experiment. Improvement of the precision of the nucleon axial form factors over a large range of momentum transfer Q^2 will be important for future precision neutrino experiments, such as DUNE. I highly recommend this paper to be published in Nature after some refinements of the manuscript and analysis:

Systematics

1) The F_A extraction depends on the inputs of other electromagnetic form factors; it would be good to demonstrate that F_A or r_A results are not changed much when different choices of form factor are used

1.a

The electromagnetic proton form factor does lead to a different proton radius in the small- Q^2 region. How does this impact the determination of r_A in the small- Q^2 region?

1.b

The F_p form factors are not so well known. There are different results, especially at small Q^2 . How would using different F_P inputs change the F_A and r_A determination?

1.c

How have the systematics associated with these choices been included in the analysis, especially for F_P ?

2) z-expansion:

A lot of parameters have been chosen to extract F_A ($t_0 = -0.75 \text{ (GeV/c)}^2$, $k_{\text{max}} = 8$, and the constraints $|ak/a_0| \lesssim 5$ and, for $k > 5$, $|ak/a_0| \lesssim 25/k$) in the F_A extraction.

2.a

There should be some explanation as to why the choices were made and how different choices might affect the final F_A determination; associated systematics should be assigned if not small.

2.b

It would be good to include a study of the stability F_A results under different t_0 , k_{max} , and other a_k constraints as an appendix. (Table 2-4 in Supplementary Material has some information but k_{max} and λ are changed at the same time.)

Updates to the manuscript

1)

Given that F_A is coupled with other form factors in the cross section, it is important to define A, B, C

2)

Refine the phrasing of the lattice comparison: "...lattice QCD, calculations are not sufficiently precise

to replace experimental input,...” LQCD results are not meant to replace experimental input, but in some cases (0.1-0.5 GeV² where there are many LQCD determinations of F_A), LQCD is as precise as experiment. For example, a recent F_A calculated directly near physical pion mass can be found in Fig. 6 of Phys.Lett.B 824 (2022) 136821

3)

Fig. 4: It may be helpful to explain why the cross-section and form-factor plots look so similar. Can one translate the black data points on the left to the right-hand plot?

4) z-expansion

4.a

Add some motivation for these constraints.

4.b

What is λ ? It's not defined in Eq. (2)

5)

In the last paragraph: It would be good to know whether F_A can be further improved (i.e. reduced errors) in the future by this or other similar experiments.

6)

Minors: Yaeggy,33 -> Yaeggy^{33}

Author Rebuttals to Initial Comments:

Referees' comments:

Color code

1. Comments that required a reply only
2. Comments that required text changes and replies
3. Comments that required additional studies, and text changes and replies
4. Comments that did not require a reply or copy editing
5. Author replies

Referee #1 (Remarks to the Author):

The paper, “First measurement of the nucleon axial vector form factor in antineutrino scattering from hydrogen” by the Minerva collaboration, discusses a measurement of the $\bar{\nu} p$ to $\mu+n$ cross section from hydrogen. This paper discusses two main points: First is the extraction of the axial form factor $F_A(Q^2)$ and second, a determination of the mean of the axial charge radius squared.

1. In the “Criteria for publication”, Nature states, “In general, to be acceptable, a paper should represent an advance in understanding likely to influence thinking in the field. . .” This is a very high mark. The abstract tells me why the transitional axial form factor is being measured this way and tells me that this “is an important input to current and future neutrino oscillation experiments”. As only an input to future experiments, I do not see the manuscript reaching this high mark. The first three paragraphs of Sec. 1 go on to explain why this is an important input to other results. The second paragraph concludes, “ $F_A(Q^2)$ is key input to the models that strongly affects the rate of charged-current scattering from nucleons.” This needs to be explained more to quantify why this measurement is likely to influence thinking in the field. These are valuable results and should be published elsewhere. I also don’t rule out that a better presentation of the importance of these results might be possible, and I could reconsider, but I do not see it now.

Reply: This result is important in three different ways.

First, it represents the first measurement of the axial vector structure of the nucleon measured using nucleons that are not bound in some nucleus. All previous probes of this structure required either some QCD theory interpretation of related processes, or required nuclear physics corrections to extract this from the bound scattering without the sort of experimental input, i.e., tags of the nuclear remnant, that would test those corrections. This is simply a form factor measurement extracted directly from the scattering cross-section.

Second, the technique we employed, measurement of neutrons in neutrino interaction in scintillator, is novel, and this technique points the way to possible future measurements in larger detectors with more intense neutrino beams.

Third, as noted by the reviewer, the result has application to other problems in neutrino physics, namely neutrino interactions that are required to measure neutrino flavor oscillations.

It is not only the last of these reasons, noted by the referee, that makes this result important. Being close to that field, we agree that we over-emphasized this last aspect in the introductory material of the original manuscript.

Text: Section 1 and the abstract have been modified to de-emphasize neutrino oscillation measurements while providing more motivation why our result is significant.

2. Does the manuscript have flaws which should prohibit its publication?

There are no serious flaws with the analysis presented in the manuscript that should prevent its publication. In fact, I am strongly in favor of its publication in any number of other journals. There were some places where more explanation would be helpful for the readers, and I will comment on some to them below.

3. For the determination of the axial form factor

a. The fit to extract the CCE signal could be better explained. In Figs. 2 and 3, my understanding is that

i. A separate fit is done for each bin in Q2.

ii. The fit is simultaneously made to three separate regions from Fig. 1. The three regions are QE, non-QE, and non-QE plus mesons.

iii. The fit uses Monte Carlo distributions of CCQE, Resonant, 2p2h, DIS, non-QELike, and CCE events.

iv. The output of the fit gives the relative normalizations of the above components. These fits are shown in Fig. 3.

v. These are then cross checked with the two validation regions to determine how good the fit really describes data in an unfitted region (not shown).

vi. Then, with the validated fit, the CCE signal is extracted from the signal region.

I know, however, that this reasoning is wrong in step (i), since only one χ^2 is quoted rather than one for each Q2 bin.

Reply: A joint fit is done for all the Q2 bins simultaneously: we calculate the χ^2 for each separate Q2 bin and then sum them together. We also add a regularization term that penalizes large differences in the weights applied to each process in adjacent Q2 bins. The strength of this regularization was chosen using the familiar “L-curve” technique. We then minimize this combined χ^2 .

Text: Updated section 10 to include a more detailed description of the fit function.

b. What are the systematic uncertainties from this fit and using the chosen regions? Did the fit take into consideration the Monte Carlo statistics or only the data statistics? (Clearly with enough computer time, the Monte Carlo statistics can be made arbitrarily large, and this question doesn't matter.)

Reply: The χ^2 of the fit is reported for the best assessment simulations, which we internally call the “central value universe”, and the χ^2 takes into account only the data

statistics. (The generated simulation's statistics is 5 times that of the data.) The statistical effect from the simulation is propagated to the signal through background subtraction.

However, we also evaluated this fit in multiple "universes", each of has variations of one of 59 different systematic uncertainties. As described in the paper, those systematic uncertainties can be broadly categorized into: final state interaction modeling for reactions on carbon, the modeling of interactions on bound nucleons in carbon, muon reconstruction uncertainties, neutrino flux uncertainty, modeling of interactions involving correlated nucleons (2p2h) uncertainty, and uncertainties related to particle responses. In assessing the final systematic uncertainty on the result, the fit result in each of these systematic "universes" is used.

This procedure then correctly accounts for both the statistical uncertainties in these fits, and the systematic uncertainty.

c. The fits shown in Fig. 2 (left) and 3 (top) have a lot of "white space" in the vertical scale that leaves me squinting to see the content.

Text: Figures are updated

d. Is "model" in Figs. 2 and 3 referring to simply the sum of the components CCQE, Resonant, 2p2h, DIS, non-QELike, and CCE events?

Reply: Yes, the model gives the Monte Carlo event rate.

e. Since the entire argument of being able to identify event types in this paper depends on the event types having substantially different distributions in the $\delta\theta_R$ vs $\delta\theta_P$ plane, it would be nice to see what these distributions are for some bin in Q2 (from Monte Carlo of course).

Reply: You can sort of getting that idea from the Q2 distribution, but here are the contour slices in angular regions in Q2, after all event selections. The color scale is arbitrary.

Text: Added these plots to the method section after Figure 6.

f. In Sec. 8, the modeling of the distribution in terms of one angular parameter with the $N(\theta)$ equation seems inconsistent with the discussion of $\delta\theta_R$ and $\delta\theta_P$ each being significant, unless it is meant to say that the A1 term corresponds to $\delta\theta_P$ and A2 corresponds to an additional widening from $\delta\theta_R$, but then there doesn't appear to be symmetry in $\delta\theta_R$, which this would imply.

Reply: $N(\theta)$ is the probability distribution of the angular resolution of the detected neutron candidate. This angular resolution is defined as the difference in angle of the detected candidate from neutron's true direction when it exits the nucleus. That angular resolution doesn't depend on which process generated the neutron.

By contrast, $\delta\theta_R$ and $\delta\theta_P$ include additional smearing from final state interactions (FSI) and Fermi motion, which are much larger effects than the neutron angular resolution.

Text: Added a sentence stating the intrinsic angular resolution is convoluted with the physics-driven neutron direction.

g. How is Fig. 6 bottom normalized? Given the abundant statistics, the histograms many standard deviations from the data.

Reply: The simulation is normalized to data. The predicted distribution that was shown is before the background fitting procedure, so the differences were due to deficiencies in the model that are corrected by this fitting.

Text: We have replaced the figure with the post-fit distribution.

4. The discussion of the axial charge radius brings up a few questions:

a. It needs to be noted that $\langle r_A^2 \rangle$ is a $\lim_{Q^2 \rightarrow 0} dFA/dQ^2$ —the limit is important—so the resolution on Q^2 and how low in Q^2 are both important. The former is available from the tables at the end of the document (although I would have liked to see it mentioned earlier). The latter, I didn't find.

Reply: $\langle r_A^2 \rangle$ is calculated at $Q^2=0$ using the fitted z-expansion description of the axial form factor that comes from our fits to the cross-section.

Text: Added text describing that the $\langle r_A^2 \rangle$ is calculated at $Q^2=0$ using the equation $[1/FA(0) dFA/dQ^2]_{Q^2=0} = -1/6 \langle r_A^2 \rangle$

b. In the discussion of the z-expansion formalism and the corresponding fit, why is $t_0 = -0.75$ (GeV/c)²? Hill and Paz, Phys. Rev. D 82, 113005 (2010) state that this is a free parameter, but also suggest that $t_0^{\text{opt}} = t_{\text{cut}} (1 - \sqrt{1 + Q_{\text{max}}^2/t_{\text{cut}}})$ which would be around -0.78 for $Q_{\text{max}} = 5.0$ GeV². The Methods Section says that t_0 was chosen “to center our precise data points around $z = 0$.” Which points are considered “precise”? From Tab. 4, these seem to be the points at larger Q^2 . Does this matter, as lower Q^2 points are closer to the $\lim_{Q^2 \rightarrow 0}$. Does this choice matter?

Reply: You are exactly right. The points we are trying to center around $z=0$ are the high Q^2 points. Changing the value of t_0 had the effect of moving the data points along the z-axis, as shown in the following figure.

The precise choice of t_0 doesn't matter, we just want to make sure the points are “balanced” on the z-plane since the expansion is a power series in z.

c. How was k_{max} chosen and was there a variation with k_{max} ? Which QCD sum rules

determine 4 parameters (perhaps this should be obvious to me)? What happens in the case of $k_{\max} = 4$, i.e, no fit—does the expansion reasonably agree with data?

Reply: We evaluated a few values of k_{\max} , from $k_{\max}=6$ to $k_{\max}=8$. At $k_{\max}=5$, the function has only 1 free parameter, similar to dipole form factor, and doesn't produce a good fit to our data. $k_{\max}=4$ has no free parameter, and is an even worse description of our data.

$k_{\max} = 8$ was chosen over $k_{\max}=6$ based on a study used to determine the regularization term (which is described in section 14 of the paper). When plotting the regularization term (penalty χ^2) against the data χ^2 , the fit at $k_{\max}=6$ shows that changing regularization has no effect on this fit, and that the chosen regularization from the L-curve study for $k_{\max}=8$ results in a lower overall χ^2 even including the regularization penalty. Therefore we used $k_{\max}=8$.

The four sum rule constraints are borrowed from Ref. 22, the Meyer et al fits to the axial form factor using the z-expansion. We asked one of the authors, Richard Hill, for the justification since it isn't explicitly given in Ref. 22, and he stated that "For a hard-scattering hadron process, every spectator quark needs to get kicked from initial to final state, so meson form factors scale as $1/Q^2$ and baryon form factors as $1/Q^4$ (formalize by light-cone wavefunctions and factorization a la Brodsky and Lepage). The power law scaling holds only up to logarithms, $\log(Q^2)$, but the nucleon form factors fall faster than $1/Q^3$." Hill suggests that the "classic reference" would be <https://inspirehep.net/literature/153236>; another review is <https://inspirehep.net/literature/194579>

Text: We have added the functional form of the four sum rule constraints into section 14 of the paper. We have added these references to the same section.

d. Why are there bounds on the values of a_k ? In particular, the fit seems to converge, according to Tab. 2, with the a_k/a_0 not near the bounds, so are they necessary?

Reply: The strength of the bound has been studied and we obtain an optimal value after scanning through values of the strength. Without the bound $k_{\max}=8$ will overfit. As noted above, the fit with $k_{\max}=6$ doesn't require the bound but the χ^2 of this fit is worse

5. If a_0 is taken from is $F_A(0) = -1.2723 \pm ?$ Is the uncertainty on this quantity meaningful in the fit? Would the Minerva data otherwise constrain $F_A(0)$?

Reply:

- 1) We tested the $\pm 1\sigma$ on $F_A(0) = -1.2723 \pm 0.0023$ and the impact on the extracted proton radius and a_k are negligible compared to the uncertainties of our measurement.
- 2) No, it is unlikely that any neutrino scattering measurement could ever be competitive in determining $F_A(0)$ compared to measurements from static properties, like the neutron lifetime.

Text: We added the uncertainty when quoting $F_A(0)$ in the end of section 1 and section 4.

6. Sec. 1, Theory, “In the limit of low negative four-momentum transfer squared, Q^2 , form factors are the Fourier transform of charge distributions.” This statement is viewed as incorrect by some theorists. See, for example, G. Miller, Phys. Rev. C 99 (2019) 3, 035202. It is a good motivational tool but needs to be acknowledged as a such.

Reply: Thank you for the comment; you are of course correct that this is a motivation to help a new reader think about the measurement. We modified the sentence and added the citation you suggested:

Text: (Form factors) “They have been thought to be the Fourier transform of charge distributions in the non-relativistic limit of low negative four-momentum transfer squared, Q^2 . Although this interpretation is not strictly true\cite{Miller:2018ybm}, the slopes of the form factors at $Q^2=0$ provide a measure of the mean squared radius $\langle r_A^2 \rangle$ for the composite particle in the charges described.”

7. Figure 6-top is critical to understand the discussion of any of the $\delta\theta^R$, $\delta\theta^P$ binning, therefore the authors should consider moving the figure to just before the discussion of Fig. 1. What is the angular resolution on the reconstructed muon track? This tells me how well the calculated neutron direction is determined.

Reply: Our muon angular resolution is on the order of 1 mrad.

Text: We relocated the equation describing the angular variables to the main text. We have not moved forward the figure to the Main text because it takes up too much space.

8. Is Fig. 1 contains a lot of information. The caption should say that the pie charts are based on the Monte Carlo fit to the data (if this is the case). The other question is how many events are in each bin in this figure?

Reply: Figure 6 in the method section gives information on how the event rate is distributed.

Text: We modified the caption to say that the figure is based on the Monte Carlo event fraction. We also added a 2D plot of $d\theta^R/P$ in Fig 1 to show the data statistics.

9. Sec. 5, p. 20, The energy distribution is as important as the central value. Knowing that it peaks at 5.5 GeV is nice, but not sufficient to calculate Q^2 . Alternatively, what is your resolution in Q^2 ?

Reply: The flux is given in the Method section and will be provided in the data release.

Here is the migration matrix for Q^2 which shows that correcting for detector resolution is straightforward. We've opted not to add it to the Method text; the Q^2 resolution of MINERvA is dominated by the muon energy resolution since Q^2 is approximately $4E_\mu(E_\mu + E_{\text{recoil}})\sin^2\theta_\mu$. That resolution is described in other MINERvA papers.

Text: We added a citation to the papers that describe the muon energy resolution in the relevant discussion in the Methods section.

10. Are the first two panels of Fig. 7 the same as are shown in Figs. 2 and 3?

Reply: Yes, they are the same plots to compare with the validation region.

Copy editing and clarity comments (no need to respond to these, just trying to be helpful):

Thank you. We have addressed all of these. There are a few comments below where the suggestion has some substantial element, or we felt it helpful to explain how we addressed it.

11. P. 5, line after Eq. 1 is not a new paragraph.

12. P. 5, is it $\langle r^2 \rangle_A$ or $\langle r_A^2 \rangle$? I would say that it is the average of the (axial radius squared) and thus the latter formulation

Text: We adopted this suggestion.

13. The variable Q_{QE}^2 is first introduced in the labeling of Figs. 2 and 3 in the main paper, but it is not defined in the text until Sec. 10 in the methods section, leaving the reader to speculate until then.

Text: I have added equations in the theory section that describe what each variable is

14. Sec. 3, p 6, last line: it would be clearer to say that the CCE signal is exclusively a subset of the QELike events?

Text: Agreed, modified the text to say “CCE is an exclusive subset of QELike”

15. Figure 5 in Sec. 2 is the first figure mentioned in the text—shouldn’t that make it Fig. 1? I must go all the way back to the Sec. 5 in Methods to see the figure. Perhaps rather than referencing the figure, just say that a more detailed description is given in the Methods section.

Text: Fixed

16. Sec. 2, p. 6, last paragraph, add a comma before the word “which”: “. . . low energy protons, which can . . .”

17. Fig. 2, p. 8: Half of the space in this figure is white space. Changing the vertical scale from $(0.0-5.0) \times 10^4$ would let the reader see details better (e.g. statistical error bars). Also, perhaps a log scale would be better so that one could see the contents of bins with $Q^2 < 8 \times 10^{-2}$?

Text: We modified the scale as suggested. We kept the linear scale because the ratio plot should provide the required information.

18. Page. 8, 2nd line, “. . . did not scatter undetectably before detection” the word “detect” in one of the two instances should be replaced; although, I understand the meaning.

Text: changed this to

“Most CCE events are in the CCE signal region, which corresponds to neutrons whose scatter was detectable.”

19. Page 10, paragraph below Eq. 3: The comma should be on the same line as “ $F_A(0) = -1.2723$ ”

20. Page 10, it is stated, “The fit at $\lambda = 0.13$ is well-behaved at the low and highest Q^2 range.” I assume that you mean that in addition to the ends, it is well behaved throughout the Q^2 range?

Text: changed to “well behaved throughout the entire Q^2 range”

21. Page 10, “The proton axial radius from the fit to this result is $r_A \equiv \sqrt{\langle r_A^2 \rangle} = 0.73(12)$ fm.” is a one sentence paragraph.

Text: Merged with previous paragraph

22. Page. 23, sentence 2. Here “Scattered particles” refer to secondary particles scattered by the (quasi-) elastically scattered neutron. In much of the rest of the paper, “scatter” refers to the primary neutron-neutrino scatter.

Text: Changed to “daughter particles”

23. P. 23, in the equation for $N(\theta)$, is $A_2 = 1 - A_1$? The use of a percentage for A_1 makes it appear so, but then why not make it explicit.

Text: Changed to “where the values of A_1 and A_2 are such that 40% of the neutrons have a narrow ...”

24. Sec. 11, p 26, paragraph 2: Do you mean analogous to Fig. 2, not 1?

Reply: Thanks for catching this mistake!

25. Don't use section headings as simply a mechanism for a new paragraph.

Text: We removed the “Control Sample” section heading.

Referee #2 (Remarks to the Author):

Dear authors,

the method presented in this paper to measure the axial proton form factor is original, effective and very promising for the future neutrino long-baseline experiments.

Congratulations for this brilliant analysis!

The data are very valuable and their reliability is assured by the long experience in running Minerva experiment since many years. The analysis is well presented.

The collaboration has large expertise in measuring cross-section of neutrino-nucleus scattering and such expertise (notably in use of statistics and treatment of uncertainties) is reliably applied in this new analysis.

Regarding the methodology and robustness of the results I have few comments below for which I would like to have your feedback. Please, do not hesitate to correct me in case of misunderstanding from my side.

1) You highlight quite strongly in the abstract the larger form factor at high Q^2 in your results with respect to previous deuterium measurement.

I find this excess pretty surprising since the deuterium corrections should impact mostly the low Q^2 region. Maybe this excess comes more from the improved functional shape (Z-expansion) than from your new method/data?

Moreover in your analysis, in this high Q^2 region, the CCE signal is much lower than the QE-like Carbon background (looking for instance at Fig.2). And even the non-QE-like background is almost as large as the CCE signal there.

As a consequence, I would like to investigate better if any residual bias is possible in the estimation of the backgrounds or elsewhere. The next few questions go in that direction.

2) One possible source of bias could be the neutron detection efficiency as a function of Q^2 in the signal.

2.a) It is pointed out indeed that in Ref.[35] a discrepancy has been observed between data and MC. I wonder if the mis-modeling of neutron secondary interactions in the detector could be dependent on the neutron energy and thus, indirectly, on the Q^2 . This would induce a wrong estimation of Q^2 -dependent efficiency correction from MC and thus, ultimately, a bias in the estimated cross-section vs Q^2 . How robust is the control of this issue?

Reply:

The discrepancy shown in Ref[35] only affects neutron interactions that produce small energy deposits. The visible protons from neutron interactions in this measurement are

required to traverse multiple strips in the detector, in order to reconstruct a 3D position, and therefore are more energetic than the neutrons in the region of the discrepancy.

Also, as described in the paper the neutron interaction model in GEANT4 at kinetic energies below 200 MeV has been tuned to match the dataset in Ref 69 and 70. At $Q_2 > 0.4 \text{ GeV}^2$, the neutrons are more energetic than this tuning supports. However, we expect interactions of high energy neutrons should be much better modeled in GEANT4 than the lower energy ones since these are predominantly inelastic, not quasielastic.

After this work, we have also studied the agreement of the neutron candidate energy distribution with our (tuned for hydrocarbon) interaction model:

This plot shows the candidate energy distribution (not the neutron true energy) in each of the Q_2 bins. We do not explicitly tune in this variable to match this dataset; rather we predict it from our tuned neutron interaction model. This plot demonstrates that the distribution is accurately predicted over the full range of neutron energies.

2.b) Another possible source of bias could be the Q_2 dependent cut of E^{recoil} (Tab1). It is my understanding that with this cut you could also affect the signal where the E^{recoil} is not a real recoil but it is due to CCE neutrons which interact already in the 100mm sphere. How much CCE signal you are cutting with Tab1 cuts and how much this cut impacts the Q_2 -dependency of the signal efficiency? If it's sizable, how you ensure that there is no-bias induced by this MC-based correction?

Reply:

It's not sizable for the signal. The Q^2 -dependent recoil energy cut is 20 MeV above the region where hydrogen events are significant. Our systematic evaluation also takes into account possible variations in particle response which could cause events to migrate to the other side of the recoil energy cut.

2.c) another possible source of bias could be the Q^2 -dependence of the muon efficiency. In the regions where large Q^2 bins are used, the efficiency for muon reconstruction may change quite a lot inside the bin and the MC-based correction is effectively an average over the assumed signal Q^2 distribution. Therefore the assumed Q^2 shape of signal in MC (if different from reality) can induce a bias in the efficiency correction and thus in the results. Could you demonstrate this effect is small?

Reply: We evaluate a CCQE shape uncertainty that changes the shape of CCQE cross-section by varying the underlying axial form factor shape. This is done using a dipole model with a wide range of axial masses from 0.9 to 1.1 (GeV/c)².

Ratios of (left) Efficiency, and (right) efficiency denominator, between systematic variations and the central value. There is a 10% change in denominator shape as a function of Q^2 . The difference in efficiency is less than 1%. The change in the slope of the ratio in the efficiency is also much less than the denominator.

(top left) Efficiency denominator ratio, (top right) efficiency numerator ratio, (bottom) efficiency ratio.

The variation created with these shifts in the axial mass, however, is not as large as the difference between the data and the dipole form factor model. To check if a larger effect as in the data is a problem, we reweighted the CCE event rate generated with the dipole cross-section to an event rate similar to data rate with a continuous and smooth weight that would move the average content in the bin as much as a plausible guess of what is in our data. The denominator contains all the generated events, while the numerator contains only the events reconstructed and selected by event selection. There is at most 3% variation in the efficiency at the highest Q^2 bin despite these larger variations.

In all cases, these variations are small compared to other systematic effects.

3) One possible source of bias could be the estimation of background. The Q^2 distribution is fit from data in control regions but, if I understand well, the extrapolation as a function of $d\theta_R$ and $d\theta_P$ (from the control region to the signal region) relies on MC simulation of neutron final state interactions and on neutron secondary interactions in the detector (and again Ref[35] pointed out a discrepancy in the simulation). Could you

comment on how well FSI and SI systematics are under control? Which systematics do you apply to FSI and SI?

Reply: We have assessed a host of FSI uncertainty within the GENIE hA model, such as the neutron mean free path (MFP) in the nucleus, neutron elastic and inelastic scattering in the nucleus, neutron absorption, charge exchange, and pion production due to neutrons in the nucleus. The largest uncertainty comes from MFP and inelastic scattering, which come out to be 3% and 1% respectively, other FSI uncertainties are sub-1% levels.

Regarding SI – as noted above, we apply a tune, and systematic uncertainties around allowed variations of that tune based on data on hydrocarbon from A. Del Guerra ([https://doi.org/10.1016/0029-554X\(76\)90181-6](https://doi.org/10.1016/0029-554X(76)90181-6)), Ref. 70. In detail, these systematic variations on the GEANT4 SI model is derived from the uncertainties in the data for the dominant neutron SI channel, i.e. $(n, n'\gamma)$, $(n, n' 3\alpha)$, $(n, n' C, B, p)$, and in the total inelastic cross-section.

The uncertainty on SI is largest for neutron kinetic energies below 100 MeV. The systematic uncertainties we assess on the SI probability are 10%~15% at these energies.

3a) I'm also worried of the patch used to fix INTRANUKE-hA: I would not expect the elastic component to have similar kinematics of no-FSI. Could you show them?

Reply: Reference [64] arXiv:1906.10576 (2019) examined the issue, based on the single-transverse kinematic imbalance (STKI) variables which are very sensitive to the FSI nucleon momentum and direction. Fig 5 of Ref [64] examines the ratio of No-FSI to the fixed elastic component in GENIE 2.12.6 in the $d\Phi_T$ variable, which is the angle between the FSI particle and the neutrino-muon plane. (It is an unsigned version of $d\Theta_P$.)

The “Elastic” in the plot is a reweighted no-FSI. The “quasielastic” has the elastic bug fixed. There is a very good agreement between the reweighted and fixed distribution. We wouldn’t be sensitive to the small variations between the model since we apply a data driven constraint. The QE fit region is 10-20 degrees surrounding the expected neutron direction and falls within the 10-20 degree region in dphiT in the above plot, which has an excellent agreement. The other fit regions cover a large dthetaP region.

3b) Sec.8 After having reweighted the GEANT4 predictions: what are the remaining assumed uncertainties on neutron SI? (Would be nice to report this into the paper as well)

Reply: As mentioned, we assumed 10~15% uncertainties on various SI channels below 100 MeV in neutron kinetic energy. We generally assume 2~4% at larger kinetic energy. The neutron SI uncertainty over the whole dataset averages to ~5%. The uncertainty is shown in Fig 11 in the method section.

Text: Added a statement on the dominant systematic uncertainties in section 4.

3c) You mention that you rely on simulation to characterize the neutron direction (with the sum of two exponential). Do you have any data control sample to confirm this?

Reply: Unfortunately, we don’t have data control over the direction. To characterize neutron direction with data requires precise knowledge of the initial neutron direction which we do not have.

We were able to check in situ the energy deposited by candidates (see our response to your comment 2a above) which is another measure of how the neutron scatters in the detector.

As also noted above, our neutron secondary interaction cross-sections are tuned to external data.

3d) The validation with neutrino events in Sec.11 does not really close successfully (large excess of events in signal region at low Q^2) and this looks worrisome to me. You mention that this could be due to large uncertainties in 2p2h but such excess is present only in the signal region (low $dT_{\theta P}$, low $dT_{\theta R}$) and not in the control region. It seems to point to a mis-modeling of the extrapolation from control to signal region which rely on neutron FSI or SI modeling (which could be present in the antineutrino events also).

Reply: The referee raised an excellent point here that our argument that this control sample was meaningful was not as convincing as it could have been. This is going to be a fairly long answer, as the suggestion led us to significantly revise the study. It's also helpful to add some text about what the test probes and what it doesn't.

The detector capabilities for finding neutrons and protons are quite different as the reconstruction is very different. One thing to note up front, is that our proton reconstruction becomes inefficient below kinetic energies of 80-100 MeV, and below the lower end of the range the efficiency is nearly zero. Q^2 is $2MT_p$ for elastic reactions, so this sets a lower bound on the Q^2 where a neutrino study can say anything about the quasielastic reaction, $Q^2 < 0.2$ (GeV/c)². In fact as the original plot in the manuscript showed, that sample becomes largely 2p2h reactions at lower Q^2 , specifically because the quasielastic ones can't be reconstructed. Similarly, the detection also isn't very efficient for high energy protons, because it relies on the proton ranging out in the detector, so the study also isn't very useful at the very highest Q^2 .

So what such a study can probe is over a limited range of Q^2 , 0.2 to ~ 2 (GeV/c)², it can verify the convolution of the initial state nuclear model and FSI. Useful for sure, since it is the same nuclear model used over all Q^2 , but not as direct at the lowest significant Q^2 measurements in this analysis.

The original study also did not make use of the best tools we have developed in our collaboration for ensuring a pure proton sample, so we also revised the analysis to adopt those. The main change is the updated proton selection (developed for arxiv: 2203.08022). This selection reduces the contribution to the sample from charged pions and reduces the number of events where the proton scatters inelastically and potentially disrupts the direction determination. We also added the full set of systematic uncertainties developed for this analysis to this study.

Here are the updated results (which will replace the results currently in the paper):

Note that there is no longer any data and very little simulation below 0.05 (GeV/c)^2 as a result of removing the backgrounds with the improved selection. In this plot, the uncertainty on MC includes both statistical and systematic errors.

The first thing to note is that the CCE region (left figure above) has a lower contribution from backgrounds and from 2p2h than the original analysis (reproduced below).

The other main thing to note is that the disagreement in the CCE region (and to a lesser extent in the QE region) is in better agreement with the model, both because the measured data/MC ratio has changed with the improved selection, but also because the full set of systematic uncertainties parallel to those in the main analysis are now included. The change in the data/MC ratio with the improved selection strongly suggests that the suppressed reactions in the new analysis, 2p2h and NonQE-like background, were the likely culprits of the larger differences in the original analysis.

The updated figure shows that there is a 10% mis-modeling in the neutrino CCE region for Q^2 between 0.2 and 0.4 (GeV/c)^2 . The disagreement is between 1-2 sigma in significance, without taking into account the trials factor, so it is a moderately significant fluctuation.

As before, our hypothesis is that varying 2p2h contribution by 100% at low Q^2 would adequately cover this disagreement. And as before, the argument is that this 100% 2p2h uncertainty is more than covered by the existing systematics in the main antineutrino analysis, since the fractional 2p2h contribution at low Q^2 is much lower because of the higher efficiency for detecting quasielastic and CCE events with the neutron in the final state.

As a complementary way of displaying the same information and bolstering our 2p2h hypothesis, in the Q^2 range where the disagreement is seen we look at the angles directly:

The CCE region sits between $-10 < \delta\theta_{P/R} < 10$. Each angle is shown at its full range and with a requirement that the other angle to satisfy the CCE selection. Disagreements in the peak vs tail for $\delta\theta_p$ and in the left side tail vs the CCE region in $\delta\theta_r$ are observed. By allowing the 2p2h to vary by $\pm 100\%$ covers those disagreements because the 2p2h (red) has a different shape than the QE contribution (green), with one exception.

The one observed difference between the data and simulation not well covered by this 100% 2p2h shift is the small asymmetry in data observed in $\delta\theta_p$. (It is not covered because the left and right tails are correlated in the 2p2h variation.) This asymmetry likely comes from the Non-QELike backgrounds [see Phys. Rev. D 100, 073010 (2019)] that are modeled to be symmetric. There are two reasons not to be concerned. First, the analysis always averages together positive and negative $\delta\theta_p$, unlike in the use of $\delta\theta_r$ where the positive and negative angles are treated differently. Second, the Non-QELike backgrounds that presumably cause the asymmetry are significantly smaller in the antineutrino measurement than in this neutrino sample.

This much improved study has replaced the previous study in the manuscript. Thank you again to the referee for inspiring us to look more carefully at this cross-check and asking us to more carefully look at our assumption about 2p2h as the origin.

Regarding the referee's comment about secondary interactions (SI), in the neutrino mode validation sample, the direction is determined by the tracked proton from the vertex, not by a line from the vertex to the point of the first interaction like in the neutron case for the main measurement. So it is difficult to really learn anything from this sample about the much larger SI effect in the signal reaction from such a study. As we have noted elsewhere, we rely on external measurements of neutron interaction on hydrocarbons and appropriate systematic uncertainties to constrain those.

Text: Updated Section 10 in the method section to contain the result of the new study. Also added a zoomed out version of the $\delta\theta_r$ in the paper to show the effect of $\pm 100\%$ shift in 2p2h and bolster the hypothesis that 2p2h is responsible for the remaining disagreement.

Further minor clarification on the analysis
 - how do you take into account the CCE component in the control regions?

Reply: The CCE component is fixed to the input prediction in the fit in the control region. This isn't a significant effect because its contribution is small.

- how large (probably small?) is the background from neutrino interactions in this antineutrino sample. What is his Q2 dependence and how well is under control?

Reply: The neutrino background is very small in this analysis. For a subset of the full simulated sample, we counted 1 neutrino event in 2470 in the QELike sample. Without any filter, we would have a distribution of the wrong-signed sample across the full Q2 space with more in the higher Q2 region, but these would produce proton tracks that would not pass our cuts. The neutrino events in the selection would fall under the various non-QELike backgrounds and be constrained alongside these backgrounds. We do not perform a separate analysis over the wrong-signed events.

- how the cuts of Table1 have been optimized?

Reply: The cut used was based on a similar approach in PhysRevD.97.052002 for MINERvA's low energy antineutrino beam run. For the medium energy beam, the recoil threshold per Q2 has been increased to accommodate more energetic interactions due to higher neutrino energies. The LE approach has a constant threshold for $Q2 < 0.16 \text{ GeV}^2$ but hydrogen events have much lower recoil energy at lower Q2. We therefore chose a lower cut that leaves a $\sim 20 \text{ MeV}$ recoil energy margin to the majority of the hydrogen events so that we could account for the migration of signal events across the recoil boundary due to uncertainties in the detector's response to different particles.

We did not use a stricter cut on the recoil energy because the fraction of signal events below the boundary is nearly constant across Q2 and recoil energy space.

- is unfolding done before or after background subtraction? Is the optimization of N_i taking into account also possible mismodeling of QELike background?

Reply: unfolding is done after background subtraction. The optimization for the number of iterations was done using fake data studies with both large variations on the signal shape and many statistical throws to ensure we don't fall into a local minimum in a significant number of pseudoexperiments.

Concerning the clarity of the presentation, I miss two important and clear explications

1) I would clearly explain the kinematics first: how do you compute Q2 (put the formula), how do you predict the expected CCE neutron direction (formula)? Explain clearly at the beginning that for CCE you have such prediction due to nucleon at rest and you use it to search the neutron in the expected direction, while the backgrounds will have different (smeared/bias) distribution around such prediction.

Text: Added the equations to be beginning.

2) Mention earlier about the measurement is convoluted with the flux

Text: Now mentioned this in both the theory section and again in the result section.

3) I find this statement a bit too strong:

“This result is the first significant measurement of the axial-vector form factor free of nuclear corrections or other theoretical assumptions”

You still have nuclear effects in the background subtraction, in the flux estimation and in the systematics on signal due to SI. I wonder what would be a better formulation for the above sentence... Maybe just acknowledging such nuclear effects explicitly right before/after that sentence would be enough...

Reply: Fair enough, although you know what we meant. Added the following line: “Theoretical uncertainties from the carbon background have been minimized by data-driven methods.”

Another major comment is about the title: here you measure the proton form factor. While the title mention the ‘nucleon’ form factor.

Reply: The form factor is an isovector transition form factor between proton and neutron. so although we measured the cross-section on the proton, it has to turn into a neutron.

Minor comments on the clarity of the text:

- Sec1: I found the following sentence not precise “past extraction have assumed FA to follow the dipole form factor, which now known not to be true for the electromagnetic factors”. I would suggest to rewrite along the lines below

Actually various attempt of using other models have been already done like in [10] but also in Eur. Phys. J. C 53, 349–354 (2008) , Phys.Rev.C 101 (2020) 2, 025501 or others. Also, I think ‘not true’ is weird wording. The dipole has always been considered a first order approximation. You may want to say that better or alternative models are now available.

Text: modified text as suggested

- The validation test with proton (instead of neutron) is mentioned in sec.3 but without any conclusion. please, put in the sec3 already if there is any left excess of events (which would be a sign of not closure of the test). If the validation works perfectly you could report just 0+/-stat uncertainty, but it is interesting to know immediately for the reader what is the level of precision of such a powerful test.

Text: Added a description of the result of the test in section 3.

- Sec 4: please list the most important systematics uncertainties here (very briefly).

Text: Added “The dominant systematic uncertainties in this analysis are the neutron secondary interaction in the detector (4.8%), the normalization in the CCQE cross-section (4.5%), the muon energy scale (4.2% from MINOS and 3.1% from MINERvA), flux (3.9%), neutron FSIs (~ 3%) and 2p2h (2.3%)” where those numbers are the uncertainty averaged across the Q2 bins of the analysis.

Please state that the theory will need, not only to apply the muon acceptance cuts and the Enu cuts, but also to convolute the prediction to the Minerva flux (and possibly a reference to it)

Text: added in Section 1 and clarified in Section 4.

- Sec.8 “linear dependency of the invariant mass”. I found this part confusing: a formula of how you predict/define θ_R would help the reader to understand.

Text: Added the equations to predict neutron direction in section 1 to illustrate the bias when the final state particle is not a neutron.

- Sec.9 “Events with a single deposit whose energy resembles proton energy deposits with at least 10 MeV per strip are defined to be QELike”. How do you distinguish such proton events from the ones where the proton is produced by the scattering of a neutron from CCE?

Reply: Energetic protons coming from the vertex will form tracks, which we remove: we require the starting point of the energy deposit to be disjoint from the vertex. There is a 20 mm cylinder around the direction of the muon track that contains the vertex. This would remove events with final state protons.

- Fig.10. In my opinion the neutron SI systematics is (with FSI) the most crucial systematics for the reliability of the analysis. It also has large impact in the intermediate Q2 region. I think it deserves its own line in the left plot of Fig.10. (And as mentioned above, I strongly suggest to add explicitly what is the assumed uncertainty after GEANT4 corrections.)

Reply: The Neutron interaction uncertainty shown in Figure 10 on the right accounts for the secondary interactions, we modified the caption to state this.

Text: “The neutron interaction systematic uncertainty accounts for the neutron secondary interaction uncertainties in the detector. The leading interaction channels, such as (n, C, Bnp) , $(n, C, 3\alpha)$, and $(n, C, n' C \gamma)$, are assigned 10% to 15% uncertainties below a kinetic energy of 100 meV.”

- Sec2: the first figure mentioned in Fig.5, why you do not put that as Fig.1 ?

Reply: Thanks for catching this! We don't have space for that figure in the main paper.

Text: we have deleted the reference that appears far from where the figure appears.

- Sec3: Fig.2 and fig3 are not mentioned anywhere in the text here (only much later)

Text: Added references to those figures in the earliest appropriate locations.

- Sec4: typo with comma on the other line after $FA(0)=-1.2723$

Text: fixed

- Sec.7: "Fermi momentum set to $K_F=221\text{MeV}$ " looks weird. Fermi momentum is a distribution, I guess you may want to say 'with maximum cutoff at $K_F=...$ '

Text: Changed to "Fermi momentum maximum of $K_F=...$ "

- Sec.8: "the the" repeated twice

Text: fixed

- Sec8: 'largely due to Fermi momentum' → "largely due to Fermi momentum for background, while for signal smearing is dominated by FSI/SI effects" Correct? I would write something like that, more explicit.

Text: Suggestion accepted

- Sec.10: "Others region" → "others region". Actually I'm not sure you defined it as 'others'. Osn't it non-QE?

Reply: It's indeed the non-QE region, thanks for catching that!

Text: It is now referred to as "non-QE and mesons".

- Sec11: " $450\text{ MeV}/c(100\text{MeV})$ " you didn't say what is the number in parenthesis (I guess kinetic energy?). Please mention it

Reply: you are correct; it is kinetic energy.

Text: fixed

Fig.8: "region 1" is not defined, probably a left-over from internal terminology

Reply: You guessed correctly. Greatly appreciated.

Text: corrected to read "QE fit region"

References

I found that some of the choices of references is not obvious. Here is a list of examples/suggestions:

- Abstract:

[1] it is weird to put only this paper. It is maybe the most recent but it for proton only (while electromagnetic form factors for neutrons are different and in this sentence you mention both protons and neutrons). A reference explaining the theory of electromagnetic form factors would look more appropriate in this sentence.

Text: Changed to Phys. Rev. 119, 1105

[9,10] I found it weird to cite [10] here where you discuss Deuterium corrections. I think

[10] apply previous Deuterium corrections from [9], it does not develop new ones. Maybe I am wrong?

Reply: You are correct.

Text: Removed [10] from the citation.

[11-13] [11] and [13] are review and they are fine but then [12] is just one single result of calculation and I guess there different results available. Why you chose that one?

Reply: [12] (Phys. Rev. D 103(3), 034509 (2021)) is the most recent result and shows that the calculation at larger Q² is not precise.

[15-19] Here you include general papers on DUNE, HK, NOVA and JUNO. Why JUNO? Is F_A relevant for JUNO? Also for T2K you cite an actual oscillation analysis results. I would quote the general T2K experiment paper (as you do for the other experiments)

Reply: JUNO was added because it would have sensitivity to atmospheric neutrinos that overlap with accelerator neutrino energies.

Text: The T2K paper has been replaced with Nucl.Instrum.Meth.A 659 (2011) 106-135

- Section1

[1] same comment as before

Reply:OK.

Text: Also added Nat Commun 12, 1759 (2021) that describes neutron radius.

[20] seems to me unnecessary

Text: Removed.

[10] and [10,24] [24] is a general review but why you include only the example of [10]? There are other papers which extracted F_A from those data (and beyond) like Phys.Rev.C 101 (2020) 2, 025501 or Eur. Phys. J. C 53, 349–354 (2008)

Text: Removed [24], added Phys.Rev.C 101 (2020) 2, 025501 and Eur. Phys. J. C 53, 349–354 (2008)

[11-13] same comment as for the abstract

Text: Changed the citation as in the abstract.

[29-31] also here I think there are other evaluations, I think. As in Phys.Rev.C 101 (2020) 2, 025501

Text: Added this reference

-Section3

[41-46] a strange selection of references what is the logic of this specific selection?

Reply: These works describe the modifications we've made to the base GENIE models.

[47] I would add also [61] here.

Text: done

- Section7

[61], [47] I would add [48] here

Text: done

- Section[8]

[39] doesn't look appropriate. Here actually we see quite relevant neutron capture for different nuclei than Carbon... It does not discuss Carbon I think.

Reply: GEANT4 evaluates neutron cross-section using data from external libraries such as ENDF(Evaluated Nuclear Data File). In the ENDF database, you can see that the radiative capture cross-section is on the order of $10^{-4} \sim 10^{-3}$ barn below a cut-off around 20 MeV. The elastic and inelastic cross-sections are on the order of 10^{-1} barn at the few MeV region.

Text: We have changed the citation to *Nuclear Data Sheets Volume 148, February 2018, Pages 1-142*

Referee #3 (Remarks to the Author):

This is a very exciting experimental result. For decades it has been very difficult to extract the nucleon axial form factors without worrying about the nuclear effect due to the large nuclei used. MINERvA Collaboration's antineutrino scattering from hydrogen is really changing the game for what we can learn about nucleon axial form factors from experiment. Improvement of the precision of the nucleon axial form factors over a large range of momentum transfer Q^2 will be important for future precision neutrino experiments, such as DUNE. I highly recommend this paper to be published in Nature after some refinements of the manuscript and analysis:

Systematics

1) The F_A extraction depends on the inputs of other electromagnetic form factors; it would be good to demonstrate that F_A or r_A results are not changed much when different choices of form factor are used

1.a

The electromagnetic proton form factor does lead to a different proton radius in the small- Q^2 region. How does this impact the determination of r_A in the small- Q^2 region?

Reply: First, using a different electromagnetic form factor has minimal impact on the determination of r_A . We studied this by changing the assumed vector form factor.

One study looked at replacing the BBBA2005 form factor with the BBBA2007 and the z-expansion electromagnetic form factor (described in PhysRevD.102.074012); the associated r_A changes by 0.01 and the uncertainty of ~ 0.17 remains the same.

We also used the results of Phys.Rev.D 102 (2020) 7, 074012, a vector form factor extraction based on the z-expansion formalism, and re-extracted F_A . At right is a plot of the ratio of F_A extracted with this alternate vector form factor ("ZexpVFF Fit") to that of BBBA2005, extracted using the methods in the paper. the same fitting parameters (k_{\max} , λ , t_0). In this case, there is a notable difference at high Q^2 , but it is still a small fraction of the overall uncertainty.

A second point is that the result is reported as a cross-section, so that anyone wishing to put in different vector form factors and re-extract an axial form factor by our prescription (or any other) could do so in the future.

1.b

The F_p form factors are not so well known. There are different results, especially at small Q^2 . How would using different F_P inputs change the F_A and r_A determination?

Reply: The pseudoscalar form factor does have an impact at low Q^2 , but its effect is largest at Q^2 below where we have precise scattering cross-sections. Phys.Rev. D86 (2012) 053003 looks (in a different context) at possible violations of the Goldberger-Treiman relation or PCAC assumptions and concluded that small variations of $F_p(0)$ and larger variations of the assumed pion pole mass in the Q^2 variation of F_p are allowed. We did a study that changes the strength of F_p by either setting it to zero or doubling it from our default assumption. The value and uncertainty of r_A were unchanged to the number of significant digits we report them.

Also, as noted above, since the result is reported as a cross-section, the axial form factor can later be re-extracted if better measurements of F_p are available in the future.

1.c

How have the systematics associated with these choices been included in the analysis, especially for F_P ?

Reply: These are very small effects in reported quantities like r_A , much smaller than other uncertainties, so we didn't include an explicit systematic uncertainty.

2) z-expansion:

A lot of parameters have been chosen to extract F_A ($t_0 = -0.75 \text{ (GeV/c)}^2$, $k_{\text{max}} = 8$, and the constraints $|a_k/a_0| \lesssim 5$ and, for $k > 5$, $|a_k/a_0| \lesssim 25/k$) in the F_A extraction.

2.a

There should be some explanation as to why the choices were made and how different choices might affect the final F_A determination; associated systematics should be assigned if not small.

2.b

It would be good to include a study of the stability F_A results under different t_0 , k_{max} , and other a_k constraints as an appendix. (Table 2-4 in Supplementary Material has some information but k_{max} and λ are changed at the same time.)

Ratio with variations of a single parameter to the central value fit. The variations are small compared to the fit uncertainty.

Fit:	Minimum Chi ²	r _A
CV Fit	9.72	0.73(17)
Alternate Fit	Change in Minimum Chi ²	r _A
t ₀ =-.55	+0.28	0.73(17)
t ₀ =-.95	0.00	0.73(16)
lambda=1	+0.53	0.72(11)
kmax=6	0.00	0.73(17)
kmax=9	0.00	0.73(17)

Reply: (2a and 2b) There are two aspects of this suggestion to respond to.

First, the selection of λ (which determines the strength of the constraint in the higher a_k) and k_{\max} is not arbitrary. It is set by the standard “L-curve” procedure described above in the response to Referee #1, 4c. Similarly, t_0 is not chosen arbitrarily; it is chosen to put the “average” of the data constraint near $z=0$ in the z -expansion, which makes sense because it is just a power law expansion in z .

Second, the effect of the small variations that are sensible given the procedures for setting λ , k_{\max} , and t_0 , are very small compared to statistical and systematic uncertainties in the analysis, as the tables and plots above show.

Third, it's not entirely clear that these belong as systematic uncertainties in the reported results given that these are not arbitrary parameters. The plot and table above show changes in the form factor, and in general, the effect is very small. Our studies showed the effects of changing these parameters have an effect below 10% of the total uncertainty, we, therefore, elect to mention them only qualitatively in the methods section.

Text: Added the following line to the end of the method section: "In general, varying the z-expansion parameters such as k_{\max} , t_0 , and λ results in changes below 10% of the total uncertainty."

Updates to the manuscript

1) Given that F_A is coupled with other form factors in the cross section, it is important to define A, B, C

Text : Added definitions of A,B,C.

2) Refine the phrasing of the lattice comparison: "...lattice QCD, calculations are not sufficiently precise to replace experimental input,..." LQCD results are not meant to replace experimental input, but in some cases (0.1-0.5 GeV² where there are many LQCD determinations of F_A), LQCD is as precise as experiment. For example, a recent F_A calculated directly near physical pion mass can be found in Fig. 6 of Phys.Lett.B 824 (2022) 136821

Text: removed the phrase "to replace experimental input"

3) Fig. 4: It may be helpful to explain why the cross-section and form-factor plots look so similar. Can one translate the black data points on the left to the right-hand plot?

Text: Added: "The resulting form factor as a ratio to the dipole form factor is shown on the right. The cross-section ratio scales approximately linearly with $\mathcal{F}\{A\}$ ratios due to the suppression of the \mathcal{A} term. "

4) z-expansion

4.a Add some motivation for these constraints.

Text: inserted citation for Phys. Rev. D 93, 113015 (2016). There is a bit more detail in the method section now; see the response to referee A, comment 4c.

4.b What is λ ? It's not defined in Eq. (2)

Reply: λ is part of the minimization function, and is now defined in eq 11 in the method section.

Text: Added more description of λ in the text.

5) In the last paragraph: It would be good to know whether F_A can be further improved (i.e. reduced errors) in the future by this or other similar experiments.

Text: Added statement on potential improvements.

6) Minors: Yaeggy,33 -> Yaeggy^{33}

Reply: It appears that the latex template has a bug that prevents multiples of 11 to appear correctly in the author list! We expect these will be fixed in the published version. Thanks for catching this!

Reviewer Reports on the First Revision:

Referees' comments:

Referee #1 (Remarks to the Author):

1. The reworking of the beginning of the paper makes a much better case than the previous version for the importance of the axial form factor on its own—thus, I would recommend publishing the paper, with a few corrections/questions.
2. In Eq. 1, the units do not work out in the numerator of E_ν . Specifically, for $M_n - M_p - m_\mu^2 + 2M_p E_\mu$, the first two terms have units of (mass) while the last two have units of (mass)².
3. In Eq. 6, taking the derivative of the first line and evaluating as $Q^2 \rightarrow 0$ does not yield the second line, specifically the factor of $1/(F_A(0))$ is extraneous.
4. The fit to extract the number of CCE events.
 - a. In the χ^2 Eq. 10, first term, the index i is summed twice, once in the overall sum (sum over S_i) and once in the Monte Carlo weighted sum in the numerator (sum over C_i). Should the latter sum be just over C ?
 - b. Does the index i refer to the bin in the QE fit region in Fig. 1? This is a sum over 12 bins in the border region of the QE signal region?
 - c. In your answer to my previous question on this fit, you said that the fit is a combined fit for all Q^2 . Is that sum implicit in the index i ?
 - d. In your answer to my previous question on Monte Carlo statistics, you replied that they were not considered in the fit, but that in general you generated a factor of 5 more Monte Carlo events than data. From Tab. 4, I see that in the first 3 bins and the last two bins, the ratio of statistical uncertainty to cross section is between 0.5 and 1.0. This implies between 1 and 3 events, or 5 to 15 Monte Carlo events. At that level Monte Carlo statistics matter, as does the Poisson nature of the data and Monte Carlo uncertainties. This ties into my observations about $\langle r_A^2 \rangle$ later.
5. The extraction of $\langle r_A^2 \rangle$ First, everything looks well within the quoted uncertainties. The article only mentions $\langle r_A^2 \rangle$ twice, (abstract and conclusions), and it is not the main point of the article. Nevertheless, given that it is mentioned,
 - a. Since the low Q^2 region is the most important, I looked at Figs. 2 (left) and 3 (top). In both plots, the data in the low Q^2 region is not visible. Would a log plot be better? Or is there nothing to be gained, given that the data is visible in the ratio plots in the accompanying plots?
 - b. In Fig. 8, left two panels, first three bins, there seems to be a systematic deviation in the data to be below the fit. How is this systematic handled?
 - c. In Figs. 3 (middle) and 4 (left) the lowest Q^2 bin is not plotted. If the paper were not making a claim about $Q^2 \rightarrow 0$ this would not bother me.

Copy editing and clarity comments (no need to respond to these, just trying to be helpful):

6. Line 122, the phrase “measured decades ago” seems to imply that old data is not good data. It’s

okay to talk about limited statistics, but age is not relevant, and dismisses many good experiments.

7. In lines 224 and 227, the muon mass is denoted as m while in line 185 and Eq. 1 it is m_μ . Both are properly explained, but consistency will help any reader.

8. Line 262, add the footnote for MINOS here, as this is important to this discussion.

9. Line 333, caption to Fig. 1, add a space between the end of the sentence and “(Right)”. In Fig. 4, the “Hydrogen [Deuterium] Fit” refer to the z-expansion fit. It might be easier for a reader if you labeled these at the z-expansion or referred to that in the caption

10. Many of the figures compare the data to the “reference model” (e. g. Figs. 2, and 3). The term is never precisely defined, which is why I asked about it last time. Personally, I would call it the result of a fit to the data. I presume that not calling it the “fit” is to distinguish this from the z-expansion presented in Sec. 4?

11. Line 901, While I know that the magnetized steel is in MINOS, the wording of this sentence is ambiguous as to whether it is in MINOS or MINERvA.

12. Figure 7, First, thank you for adding these plots. They do an excellent job of explaining the difference in the distributions for the various types of events. They have, however, vastly different color scales, ranging from yellow = 540 down to yellow = 18—a factor of 30. That difference needs to be called out in the caption, or perhaps a log scale on the color? (I’m not sure if that would be better though.)

13. Figure 8, the middle plot, “QE fit region” is from Fig. 3, not Fig. 2.

14. Figure 12 (left) Does this figure present any information that is not in Fig. 4 if I multiply by the dipole form factor? That is fine, but I wanted to make sure I understand

15. Figure 12 (right) it might also be useful to mention this in the first paragraph of Sec. 5, where the beam is described.

16. Table 2, Why are the uncertainties only shown for some of the a_i parameters but not all?

17. Table 4, none of the uncertainties are super or subscripted as mentioned in the caption. I assume at one time asymmetrical uncertainties were quoted, but then dropped without changing the caption.

Referee #2 (Remarks to the Author):

Dear authors,

thanks for this improved version of the document. I really appreciate the effort to improve clarity and readability of such an important result.

Let me stress again that this is obviously an important measurement by itself but it is also opening the road to a new rich set of measurements using neutrons in these type of detectors.

I have few remaining questions on the analysis and few minor comments on the text.

Questions on analysis procedure

I understand that 2p2h and CCQE normalization are left free and constrained by the fit to the data in control regions. So I do not understand why you have (L461-464) remaining few percent systematics uncertainties due to their cross-sections. Is this an additional shape systematics on the $\Delta\theta_{P,R}$ distribution?

On a similar note, I understand that 2p2h normalization is left free and constrained by the fit to the data in control regions separately in the validation with neutrino sample.

If that is the case then I do not understand why you mention an additional 100% uncertainty on 2p2h (L1209-1210 and Fig9)

To assess the goodness of the validation in the validation regions (QE-like and nonQE-like) the reader need to see the actual event distributions (data vs tuned model after fit to the control region).

I do not think this plot is included. I can see only the plot of Fig.8 (right) but I think here you compare to the reference model so we cannot assess how good is the closure test in the validation region.

I would suggest to substitute this plot with a comparison between the data and the tuned/postfit model both for the QE-like and nonQE-like validation regions.

L1162-1163: I do not understand how you use λ_S , if this parameter is used to adjust the validation regions more details need to be added on the level of tuning needed to ensure consistency

L1272-1273: I do not understand this sentence "The analog to the CCE selection selects the two right most non-zero bins."

L1305: why to test the unfolding you decided to use a toy model which differs only in the high Q2 region?

Minor comments on the text

L151-152 it is a bit weird to mention only mass hierarchy and CP violation, especially considering that the systematics on F_A is actually more relevant for precision measurements of mixing angles and mass differences.

I would suggest a sentence like "... to precisely measure the neutrino oscillation parameters,

including the strength of CP violation, and to establish mass hierarchy"
(indeed, it is also weird to "precisely measure" mass hierarchy, which is a discrete parameter)

L300-301

I think saying that it is 'impossible' to extract F_A directly from CCQE in C is a bit too strong. I would say that any attempt is undermined by very large uncertainties on nuclear physics, or something like that...

Fig6 colored distributions are "after sideband fits" (in the caption) but L1028 says that these are "reference model distributions". I think this is a typo right?

I think in the rest of the paper (eg Fig2,3) you use "reference model" to define the MC before tuning from control regions.

L1157 typo: fititing -> fitting

Fig11: I would strongly suggest to zoom the y axis of Fig.11 (right) to make visible the various contributions: it is the first time an analysis of this type is conducted but I expect a bright future for the measurement of neutrons in such kind of experiments/detector so it would be very interesting for the community to know what are the most relevant systematics for neutron detection, notably for the simulation of neutron interactions.

You may mention in the caption that the y-axis scales are different between left and right figure.

Referee #3 (Remarks to the Author):

I am happy to see the authors have answered many of my previous concerns. I am satisfied with their replies except for the few items below.

On the FP-dependent study,

1) Please include a comparison of F_A as a function of Q^2 (and the resulting r_A) with different inputs of FP in the appendix. This would be valuable, because the claimed lack of dependence on the form of FP is not intuitive.

2) I missed this in the first report "The pseudoscalar form factor is predicted from the axial form factor[31]."

"[31] Goldberger, M.L., Treiman, S.B.: Decay of the pi meson. Phys. Rev. 110, 1178–1184 (1958). <https://doi.org/10.1103/PhysRev.110.1178>"

Are there more modern inputs one can use? For example, there are lattice calculations of the pseudoscalar form factors. The authors should include lattice pseudoscalar form factors and assess their impact on the results.

In the abstract:

"Calculations of F_A from lattice QCD are underway, but are not yet precise[10–12]."

And "Theory" section:

"While many efforts[10–12] are underway to calculate F_A for $Q^2 > 0$ from lattice QCD,

calculations are not sufficiently precise”

1) In the previous report I noted that “LQCD results are not meant to replace experimental input, but in some cases (0.1-0.5 GeV² where there are many LQCD determinations of F_A), LQCD is as precise as experiment. For example, a recent F_A calculated directly near physical pion mass can be found in Fig. 6 of Phys.Lett.B 824 (2022) 136821”, so the authors’ sentence is not correct. Can the authors update the text to reflect the facts?

2) There are many missing lattice references on F_A calculations in physical pion mass. The selection of references used here does not reflect the complete lattice-community efforts in F_A . The authors should at least have the references from the caption of Fig. 6 of Phys.Lett.B 824 (2022) 136821”, including that paper itself (which uses completely different methods to obtain F_A , in contrast to previous lattice works).

3) A plot comparing with lattice F_A should be included.

With these updates, the manuscript is ready for publication.

Author Rebuttals to First Revision:

Color code

1. Comments that required a reply only
2. Comments that required text changes and replies
3. Comments that required additional studies, and text changes and replies
4. Comments that did not require a reply, or copy editing
5. Author replies

Note: We have realized the event ratio plots contain a bug that double counts the MC uncertainty in the plots, we've corrected the error and the plots in the paper are updated. The event rate plots are not affected.

Referee #1 (Remarks to the Author):

1. The reworking of the beginning of the paper makes a much better case than the previous version for the importance of the axial form factor on its own—thus, I would recommend publishing the paper, with a few corrections/questions.

2. In Eq. 1, the units do not work out in the numerator of E_v . Specifically, for $M_n - M_p - m_\mu^2 + 2M_p E_\mu$, the first two terms have units of (mass) while the last two have units of (mass)².

Reply: Thank you for catching this typo – we missed a power of 2 on the first two terms and they have been added.

3. In Eq. 6, taking the derivative of the first line and evaluating as $Q^2 \rightarrow 0$ does not yield the second line, specifically the factor of $1/(F_A(0))$ is extraneous.

Reply: This is another typo; $FA(0)$ should be a factor multiplying the series expansion, we've corrected it – $FA(Q^2) = FA(0)(1 - \dots)$

4. The fit to extract the number of CCE events.

a. In the χ^2 Eq. 10, first term, the index i is summed twice, once in the overall sum (sum over S, i) and once in the Monte Carlo weighted sum in the numerator (sum over C, i). Should the latter sum be just over C ?

Reply: Yes, the summation should be over C and not (C, i)

b. Does the index i refer to the bin in the QE fit region in Fig. 1? This is a sum over 12 bins in the border region of the QE signal region?

Reply: The index i refers to the i th Q^2 bin. C is the MC event type category, i.e. CCQE, 2p2h, resonant, etc, while S is designated for the different angular regions.

Text: We've updated the text immediately following eq 10 to make these designations more clear.

c. In your answer to my previous question on this fit, you said that the fit is a combined fit for all Q^2 . Is that sum implicit in the index i ?

Reply: yes, the summation over index i combines the χ^2 for all Q^2 bins.

d. In your answer to my previous question on Monte Carlo statistics, you replied that they were not considered in the fit, but that in general you generated a factor of 5 more Monte Carlo events than data. From Tab. 4, I see that in the first 3 bins and the last two bins, the

ratio of statistical uncertainty to cross section is between 0.5 and 1.0. This implies between 1 and 3 events, or 5 to 15 Monte Carlo events. At that level Monte Carlo statistics matter, as does the Poisson nature of the data and Monte Carlo uncertainties. This ties into my observations about $\langle r_A^2 \rangle$ later.

Reply:

The bins between (0,0.0125) shown in figures 2 and 3 are not reported in Table 5. Similarly, the last bin (6.0,10), is not reported. Before unfolding, the first reported bin contains 7.6 events after background subtraction and the last reported bin contains 17 events. The statistical uncertainties in these bins must account for the background uncertainty when we do the subtraction, and therefore they are larger than the square root N uncertainty from the event rate alone. The MC uncertainty would also come into play during the efficiency correction stage which will increase the total uncertainty. So yes, the reviewer is correct that the MC statistics have caused a pronounced increase in the statistical uncertainty of the data at the low Q2 region.

Below is a table with the numbers of events in the CCE region and the predicted background. The first two bins have very few events. In the bins with a reasonably precise measurement that is important for the fits, there are hundreds of events. Even in the imprecise first three reported bins below 0.05 GeV² and the bin from 4-6 GeV², the numbers of events are more than ten, which should mitigate the concern about the significance of the non-Gaussian Poisson fluctuations?

Bin Low Edge	Bin High Edge	Measured Events	Predicted Background Events	Statistical Uncertainty on Measured Events after Background Subtraction
0	0.00625	0	0	0
0.00625	0.0125	1	0	1
0.0125	0.025	11	3.3	3.6
0.025	0.0375	18	8.3	4.6
0.0375	0.05	31	22.2	7.0
0.05	0.1	328	182	233
0.1	0.15	684	372.0	32.9
0.15	0.2	1011	523.7	39.1
0.2	0.3	2507	1401	65
0.3	0.4	2485	1563	66
0.4	0.6	3908	2861	84
0.6	0.8	2540	1919	69
0.8	1	1604	1262	55
1	1.2	1019	788.3	42.0

1.2	2	1494	1164	53
2	4	395	289	25
4	6	37	20.6	6.8
6	10	8	4.4	3.1

Note that the Monte Carlo statistical error is not simply \sqrt{N} because the events are weighted, including a weight accounting for the factor of five more statistics in Monte Carlo. This table has been added as Table 4 in the appendix.

5. The extraction of $\langle r_A^2 \rangle$ First, everything looks well within the quoted uncertainties. The article only mentions $\langle r_A^2 \rangle$ twice, (abstract and conclusions), and it is not the main point of the article. Nevertheless, given that it is mentioned,

a. Since the low Q^2 region is the most important, I looked at Figs. 2 (left) and 3 (top). In both plots, the data in the low Q^2 region is not visible. Would a log plot be better? Or is there nothing to be gained, given that the data is visible in the ratio plots in the accompanying plots?

Reply: We don't think a log plot would provide more information than the ratio plot. Note that the lowest Q^2 bins are not reported as a cross-section result or used in the fit as described above in response to 4d. We also provided a table containing the raw data rate and fitted MC event rate for all Q^2 bins, with their respective statistical uncertainties, in the appendix.

b. In Fig. 8, left two panels, first three bins, there seems to be a systematic deviation in the data to be below the fit. How is this systematic handled?

Reply: We did not assign separate systematics for the low Q^2 bins because the data and MC are consistent within uncertainties. Note that the systematic uncertainties between adjacent bins in the region of that deviation are highly correlated.

c. In Figs. 3 (middle) and 4 (left) the lowest Q^2 bin is not plotted. If the paper were not making a claim about $Q^2 \rightarrow 0$ this would not bother me.

Reply: The strategy used to extract the axial radius is simply calculating it from the fit. This fit also makes minimal assumptions other than the high- Q^2 behavior and the normalization at $Q^2=0$ through electro-production of pions. Therefore we don't make a claim about extracting the radius directly from data (we simply can't because our technique suffers from poor efficiency at low Q^2). Because the data is fit to the z-expansion, it is not intuitive how any single data point contributes to the uncertainty in the slope at zero Q^2 .

Copy editing and clarity comments (no need to respond to these, just trying to be helpful):

6. Line 122, the phrase "measured decades ago" seems to imply that old data is not good data. It's okay to talk about limited statistics, but age is not relevant, and dismisses many good experiments.

Reply: We have rewritten the abstract to remove "decades ago" and instead highlight that we report the first statistically significant measurement of the CCE process.

7. In lines 224 and 227, the muon mass is denoted as m while in line 185 and Eq. 1 it is m_μ . Both are properly explained, but consistency will help any reader.

Reply: Equations 4 and 5 are generally applicable for charged leptons. We've replaced "muon mass" to be "charged lepton mass".

8. Line 262, add the footnote for MINOS here, as this is important to this discussion.

Reply: Added. Previously this was Ref 55, but we realized also that this was a recent, but not comprehensive detector description and so have changed the paper we cited from MINOS.

9. Line 333, caption to Fig. 1, add a space between the end of the sentence and “(Right)”. In Fig. 4, the “Hydrogen [Deuterium] Fit” refer to the z-expansion fit. It might be easier for a reader if you labeled these at the z-expansion or referred to that in the caption

Reply:Added reference in the caption, thanks!

10. Many of the figures compare the data to the “reference model” (e. g. Figs. 2, and 3). The term is never precisely defined, which is why I asked about it last time. Personally, I would call it the result of a fit to the data. I presume that not calling it the “fit” is to distinguish this from the z-expansion presented in Sec. 4?

Reply: You are right. We have changed the language, in many places, to be “post-fit model”. The models from tunes performed by MINERvA, before the fit, are still referred to as the “reference model”.

11. Line 901, While I know that the magnetized steel is in MINOS, the wording of this sentence is ambiguous as to whether it is in MINOS or MINERvA.

Reply: Added “... steel in MINOS ...”

12. Figure 7, First, thank you for adding these plots. They do an excellent job of explaining the difference in the distributions for the various types of events. They have, however, vastly different color scales, ranging from yellow = 540 down to yellow = 18—a factor of 30. That difference needs to be called out in the caption, or perhaps a log scale on the color? (I’m not sure if that would be better though.)

Reply: We tried different combinations and this is the clearest we were able to do.

Text: We added a sentence saying the scale is different for each event type due to the event rate, into the caption.

13. Figure 8, the middle plot, “QE fit region” is from Fig. 3, not Fig. 2.

Reply: We did mention QE fit is from Fig. 3. For clarity, the modified text reads: “CCE and QE fit regions are reproduced from Fig. 2 and Fig. 3, respectively, for comparison”

14. Figure 12 (left) Does this figure present any information that is not in Fig. 4 if I multiply by the dipole form factor? That is fine, but I wanted to make sure I understand

Reply: There is no new information, but we thought it would be useful for the reader.

15. Figure 12 (right) it might also be useful to mention this in the first paragraph of Sec. 5, where the beam is described.

Reply: Added a reference to this figure.

16. Table 2, Why are the uncertainties only shown for some of the a_i parameters but not all?

Reply: Only those a_i fitted have independent uncertainties. For example, for $k_{max}=8$, only $a_{\{1,2,3,4\}}$ are fitted, the rest of the a_j parameters are calculated from the values of $a_{\{1,2,3,4\}}$ using the constraints.

17. Table 4, none of the uncertainties are super or subscripted as mentioned in the caption. I assume at one time asymmetrical uncertainties were quoted, but then dropped without changing the caption.

Reply: Thanks for the catch!

Referee #2 (Remarks to the Author):

Dear authors,

thanks for this improved version of the document. I really appreciate the effort to improve clarity and readability of such an important result.

Let me stress again that this is obviously an important measurement by itself but it is also opening the road to a new rich set of measurements using neutrons in these type of detectors.

I have few remaining questions on the analysis and few minor comments on the text.

Questions on analysis procedure

I understand that 2p2h and CCQE normalization are left free and constrained by the fit to the data in control regions. So I do not understand why you have (L461-464) remaining few percent systematics uncertainties due to their cross-sections. Is this an additional shape systematics on the $\Delta\theta_{P,R}$ distribution?

Reply:

The fit does not have enough freedom to perfectly describe the data in the sub-samples that are being used in the fit because it is scaling sub-components with the additional constraint of a regularization to suppress large bin-to-bin discontinuities in the scale factors. The residual difference between the data and the post-fit model that remains because of this is different in different systematic variations that are considered to evaluate that uncertainty. This results, mechanically, in a “systematic uncertainty” associated with these differences. We believe this is appropriate to include as a systematic uncertainty in the final result. Added the following text to be helpful (L462):

Text: Systematic uncertainties arise from the small remaining differences, due in part to the regularization, between the post-fit background prediction in each systematic variation of the input model.

On a similar note, I understand that 2p2h normalization is left free and constrained by the fit to the data in control regions separately in the validation with neutrino sample.

If that is the case then I do not understand why you mention an additional 100% uncertainty on 2p2h (L1209-1210 and Fig9)

Reply: The validation test reported here is done using **neutrino** mode CCQE, with a proton final state. The discrepancy in this mode indicates a 100% uncertainty on the 2p2h within a restricted phase space, however, the uncertainty on our **antineutrino** data adequately covers that. The 100% is meant as an upper bound on the uncertainty from this source.

To assess the goodness of the validation in the validation regions (QE-like and nonQE-like) the reader need to see the actual event distributions (data vs tuned model after fit to the control region).

I do not think this plot is included. I can see only the plot of Fig.8 (right) but I think here you compare to the reference model so we cannot assess how good is the closure test in the validation region.

I would suggest to substitute this plot with a comparison between the data and the tuned/postfit model both for the QE-like and nonQE-like validation regions.

Reply: We have replaced Fig. 8 with the QE and Non-QE validation regions plots. Please note that mechanically, the Non-QE validation has been separated into two sub-regions that have no impact on how we assess the goodness of the fit. As the other fit regions are displayed in the main section, we've elected to only show the validation regions here for a coherent presentation.

L1162-1163: I do not understand how you use λ_S , if this parameter is used to adjust the validation regions more details need to be added on the level of tuning needed to ensure consistency

Reply: We added new text to describe the procedure

Text: To do so, the fit is performed on the central value model for a range of λ_S and an average χ^2 for the validation regions, weighted by their event rates, is calculated. The value of λ_S is chosen through an L-curve study.

L1272-1273: I do not understand this sentence "The analog to the CCE selection selects the two right most non-zero bins."

Reply: By "two right most non-zero bins" we meant the bins between -10 and 10 degrees. But now we realize this statement is not correct since the bin at 60 degrees has small but non-zero content.

Text: The analog to the CCE selection selects between -10 degree to 10 degree.

L1305: why to test the unfolding you decided to use a toy model which differs only in the high Q2 region?

Reply: The model actually differs from our reference model across the entire Q2 range, but the high Q2 region is mentioned explicitly because the difference larger there than in the low Q2 region.

Modified Text: The toy model differs from the reference model by about 10% at lower Q^2 , and 20% at higher Q^2 , in order to mimic the shape of the data model discrepancy.

Minor comments on the text

L151-152 it is a bit weird to mention only mass hierarchy and CP violation, especially considering that the systematics on F_A is actually more relevant for precision measurements of mixing angles and mass differences.

I would suggest a sentence like "... to precisely measure the neutrino oscillation parameters, including the strength of CP violation, and to establish mass hierarchy" (indeed, it is also weird to "precisely measure" mass hierarchy, which is a discrete parameter)

Reply: Thanks for the suggestion. We updated the sentence.

L300-301

I think saying that it is 'impossible' to extract F_A directly from CCQE in C is a bit too strong.

I would say that any attempt is undermined by very large uncertainties on nuclear physics, or something like that...

Reply: Updated the sentence to read:

Text: While the CCQE cross-section is a function of F_a as well, measurements of electromagnetic form factors even in nuclei as light as ^4He have shown that nuclear effects obscure the relationship and make extracting F_a directly from CCQE in ^{12}C susceptible to uncertain nuclear physics.

Fig6 colored distributions are "after sideband fits" (in the caption) but L1028 says that these are "reference model distributions". I think this is a typo right?

I think in the rest of the paper (eg Fig2,3) you use "reference model" to define the MC before tuning from control regions.

Reply: Good catch! We updated the figure during the previous comment period and missed this. Note that there is a related comment from Referee #1 about our nomenclature, and that in response we are consistently using "reference model" for pre-fit and "post-fit model" for the result after the fit.

L1157 typo: fititing -> fitting

Reply: Thanks

Fig11: I would strongly suggest to zoom the y axis of Fig.11 (right) to make visible the various contributions: it is the first time an analysis of this type is conducted but I expect a bright future for the measurement of neutrons in such kind of experiments/detector so it would be very interesting for the community to know what are the most relevant systematics for neutron detection, notably for the simulation of neutron interactions. You may mention in the caption that the y-axis scales are different between left and right figure.

Reply: We have added the systematic uncertainties on a log y scale.

Referee #3 (Remarks to the Author):

I am happy to see the authors have answered many of my previous concerns. I am satisfied with their replies except for the few items below.

On the FP-dependent study,

1) Please include a comparison of FA as a function of Q2 (and the resulting r_A) with different inputs of FP in the appendix. This would be valuable, because the claimed lack of dependence on the form of FP is not intuitive.

$$A = \frac{(m^2 - q^2)}{4M^2} \left[\left(4 - \frac{q^2}{M^2}\right) |F_A|^2 - \left(4 + \frac{q^2}{M^2}\right) |F_V^1|^2 - \frac{q^2}{M^2} |\xi F_V^2|^2 \left(1 + \frac{q^2}{4M^2}\right) - \frac{4q^2 \operatorname{Re} F_V^1 \xi F_V^2}{M^2} \right. \\ \left. + \frac{q^2}{M^2} \left(4 - \frac{q^2}{M^2}\right) |F_A^3|^2 - \frac{m^2}{M^2} \left(|F_V^1 + \xi F_V^2|^2 + |F_A + 2F_P|^2 + \left(\frac{q^2}{M^2} - 4\right) \left(|F_V^3|^2 + |F_P|^2\right)\right) \right]$$

$$F_P(Q^2) = \frac{2M^2 F_A(Q^2)}{M_\pi^2 + Q^2}.$$

Reply: Here are the ratios of the z-expansion form factor that is derived with different strengths of Fp. The effect is very small.

The largest effect of the pseudoscalar form factor is at values of $Q^2 \sim m_\pi^2$, since the induced pseudoscalar form factor from the Goldberger-Treiman relation is $FA^*(2M^2/(m_\pi^2 + Q^2))$. Our lowest Q^2 at which we have a reasonably precise measurement of the cross-section is at $Q^2 \sim 0.05 \gg m_\pi^2$, so for simplicity we can consider this to be $FA^*(2M^2/Q^2)$. There is also a $m^2/M^2 = (m_{\text{lepton}}/m_{\text{nucleon}})^2 \sim 1\%$ suppression of all terms involving Fp in the cross-section.

The only contribution is to the "A" term of Eqns 4-5, and the contributions are $(m/M)^2[-4FA^*Fp + Q^2Fp^2/M^2]$ which is approximately $(m/M)^2*(FA^2)*(-8M^2/Q^2 + 4M^2/Q^2)$ or $-4(m^2/Q^2)FA^2$. This is to be compared to the

leading FA dependence in the cross-section from the C term which, for $ME_{\nu} \gg Q^2$, which again is true for the precise data in this measurement, the leading term in the cross-section proportional to FA^2 is $4(E_{\nu}^2/M^2)FA^2$, again leading to the conclusion that the induced FP contribution should be a small contribution, a few % at most, in our high energy flux integrated cross-section for $Q^2 \gg m_{\pi}^2$.

2) I missed this in the first report “The pseudoscalar form factor is predicted from the axial form factor[31].”

“[31] Goldberger, M.L., Treiman, S.B.: Decay of the pi meson. Phys. Rev. 110, 1178–1184 (1958). <https://doi.org/10.1103/PhysRev.110.1178>”

Are there more modern inputs one can use? For example, there are lattice calculations of the pseudoscalar form factors. The authors should include lattice pseudoscalar form factors and assess their impact on the results.

Reply: This older formulation of FP has been chosen because the experimental neutrino community still uses it and would allow this audience to better interpret our result. We agree there are probably newer formulations, but as we have shown, extreme variations of assumptions on the induced FP do not matter. The data is also available for anyone to fit to with any model of FA, even following the procedures that we have spelled out in the paper.

In the abstract:

“Calculations of FA from lattice QCD are underway, but are not yet precise[10–12].”

And “Theory” section:

“While many efforts[10–12] are underway to calculate FA for $Q^2 > 0$ from lattice QCD, calculations are not sufficiently precise”

Reply: The main text has been updated as follows:

Text: While there are many efforts to calculate Fa for $Q^2 > 0$ from lattice QCD with increasing precision, calculations in the Q^2 region above 1GeV^2 remain imprecise.

1) In the previous report I noted that “LQCD results are not meant to replace experimental input, but in some cases ($0.1\text{--}0.5\text{ GeV}^2$ where there are many LQCD determinations of F_A), LQCD is as precise as experiment. For example, a recent F_A calculated directly near physical pion mass can be found in Fig. 6 of Phys.Lett.B 824 (2022) 136821”, so the authors’ sentence is not correct. Can the authors update the text to reflect the facts?

Reply: We updated the abstract and included Phys.Lett.B 824 as a citation.

Text: The antineutrino-hydrogen scattering presented here can access the axial form factor without the need for nuclear theory, and allows direct comparisons with the increasingly precise lattice QCD computations[11–14].

2) There are many missing lattice references on FA calculations in physical pion mass. The selection of references used here does not reflect the complete lattice-community efforts in FA. The authors should at least have the references from the caption of Fig. 6 of Phys.Lett.B 824 (2022) 136821”, including that paper itself (which uses completely different methods to obtain FA, in contrast to previous lattice works).

Reply: We are citing a review because we are near the citation limit. We added Phys.Lett.B 824 but may not be able to add any more citations.

3) A plot comparing with lattice F_A should be included.

Reply: We updated Fig. 4 (right) with a ratio of the Phys.Lett.B 824 fit to the dipole form, labeled “LQCD fit”.

With these updates, the manuscript is ready for publication.

Reply: Thank you!

Reviewer Reports on the Second Revision:

Referees' comments:

Referee #1 (Remarks to the Author):

This (the 3rd) manuscript, I feel is ready for publication. The authors have answered my main questions or clarified my misunderstandings of the text.

There are a few things which the authors might want to consider before publication, but it is not necessary that their response would need review again.

1. In Tab. 2 of the Supplementary Material, perhaps say in the caption that the a_k parameters given without uncertainties are not free variables in the fit but set by the sum rule in Eq. 13. Sorry I missed this in the previous version—which is why I would point it out.
2. Table 4, I know that I can do the math, but you might want to add a column, since there is room, of events – background.
3. Table 4 caption "Only bins 3-27 is. . ." change to "Only bins 3-27 are. . ."
4. Table 4, is the uncertainty on 0 measured events with 0 expected really 0?

Referee #2 (Remarks to the Author):

Dear authors,

thanks for the answers and for the improvements to the text/figures which you implemented. I'm happy with the present version of the document to be published.

Let me congratulate again with you for this milestone result!

Referee #3 (Remarks to the Author):

I am happy with the updates. I recommend the paper to be published.

Referee Response

Thank you again to all the referees for their work to improve our paper. We are truly grateful.

Referee #1 (Remarks to the Author):

This (the 3rd) manuscript, I feel is ready for publication. The authors have answered my main questions or clarified my misunderstandings of the text.

There are a few things which the authors might want to consider before publication, but it is not necessary that their response would need review again.

1. In Tab. 2 of the Supplementary Material, perhaps say in the caption that the a_k parameters given without uncertainties are not free variables in the fit but set by the sum rule in Eq. 13. Sorry I missed this in the previous version—which is why I would point it out.

Reply: We think this is a good suggestion, and have modified the caption as such

2. Table 4, I know that I can do the math, but you might want to add a column, since there is room, of events – background.

Reply: We added a new column to show the background subtracted rate.

3. Table 4 caption "Only bins 3-27 is. . ." change to "Only bins 3-27 are. . ."

Reply: Thanks for spotting it.

4. Table 4, is the uncertainty on 0 measured events with 0 expected really 0?

Reply: We dropped the uncertainty in that bin in the table. Within the limitations of our simulated statistics, we are unable to identify any events, and therefore are unable to estimate a non-zero prediction for the process. It does not matter because we do not report this bin.

Referee #2 (Remarks to the Author):

Dear authors,

thanks for the answers and for the improvements to the text/figures which you implemented. I'm happy with the present version of the document to be published.

Let me congratulate again with you for this milestone result!

Referee #3 (Remarks to the Author):

I am happy with the updates. I recommend the paper to be published.